| **Open Peer Review** | Microbial Ecology | Methods and Protocols

# Streamlined extraction of nucleic acids and metabolites from low- and high-biomass samples using isopropanol and matrix tubes

Caitriona Brennan,[1,2] Justin P. Shaffer,[3] Pedro Belda-Ferre,[1] Ipsita Mohanty,[4] Yuhan Weng,[1,5] Kalen Cantrell,[5] Gail Ackermann,[1] Celeste Allaband,[1] MacKenzie Bryant,[1] Sawyer Farmer,[1] Antonio González,[1] Daniel McDonald,[1] Cameron Martino,[1] Michael J. Meehan,[4] Gibraan Rahman,[1] Rodolfo A. Salido,[1,6] Tara Schwartz,[1] Se Jin Song,[7] Caitlin Tribelhorn,[1] Helena M. Tubb,[1] Pieter C. Dorrestein,[1,4,7] Rob Knight[1,6,7,8]

**ABSTRACT** An essential aspect of population-based research is collecting samples outside of a clinical setting. This is crucial because microbial populations are highly dynamic, varying significantly across hosts, environments, and time points, a variability that clinical sample collection alone cannot fully capture. At-home sample collection enables the inclusion of a larger and more diverse group of participants, accounting for differences in ethnicity, age, and other factors. However, managing large studies is challenging due to the complexities involved in sample acquisition, processing, and analysis. Building on our previous work demonstrating the effectiveness of single 1 mL barcoded, racked Matrix Tubes in reducing sample processing time and well-to-well contamination for paired DNA and metabolite extraction, we further validate this method against a previously benchmarked plate-based approach using the same extraction reagents. This validation focuses on samples from the built environment, human skin, human saliva, and feces from mice and humans. Importantly, we explore the impact of using a mix of bead sizes during bead-beating for cell lysis, demonstrating that it enhances taxonomic recovery compared to a single bead size. Finally, we assess the potential of 95% isopropanol for room-temperature sample preservation. Our results show that isopropanol performs comparably to 95% ethanol in many cases, suggesting it is viable as an alternative when ethanol is unavailable. Beyond minimizing contamination, halving processing time, eliminating human error during sample plating, and streamlining metadata curation, the Matrix tube approach produces metabolomic, 16S, and shotgun metagenomic data consistent with the Plate-based Method for both high- and low-biomass samples.

**IMPORTANCE** Numerous studies have linked the microbiome to human and environmental health, yet many fundamental questions remain unanswered. Large-scale studies with robust statistical power are required to identify important covariates against a background of confounding factors. Cross-contamination, limited throughput, and human error have been identified as major setbacks when processing large numbers of samples. We present a streamlined method for sample accession and extraction of metabolites and DNA for both high- and low-biomass samples. This approach, previously shown to significantly reduce cross-contamination, employs an automation-friendly, single barcoded tube per sample. Additionally, we demonstrate that 95% isopropanol serves as an effective ambient-temperature storage solution for many sample types, providing an alternative in regions where ethanol is unavailable or restricted. This method has significant implications for the field, enabling large-scale studies to generate

Address correspondence to Rob Knight, rknight@ucsd.edu.

Caitriona Brennan and Justin P. Shaffer contributed equally to this article. The author order was determined alphabetically.

R.K. is a scientific advisory board member and consultant for BiomeSense, Inc., has equity, and receives income. He is a scientific advisory board member and has equity in GenCirq. He has equity in and acts as a consultant for Cybele. He is a co-founder of Biota, Inc. and has equity. He is a co-founder of Micronoma and has equity and is a scientific advisory board member. He is a board member of Microbiota Vault, Inc. He is a board member of N=1 IBS advisory board and receives income. He is a Senior Visiting Fellow of HKUST Jockey Club Institute for Advanced Study. D.M. is a consultant for, and has stock in, BiomeSense, Inc. P.C.D. is an advisor and holds equity in Cybele, Sirenas, and BileOmix and is a scientific co-founder and advisor for and holds equity in Ometa, Enveda, and Arome with prior approval by University of California San Diego. P.C.D. also consulted for DSM animal health in 2023. M.J.M. is a full-time employee of Thermo Fisher Scientific and receives income. P.B.-F. is a full-time employee of Element Biosciences, has equity, and receives income. The terms of these arrangements have been reviewed and approved by the University of California San Diego, in accordance with its conflict of interest policies.

See the funding table on p. 25.

accurate insights with greater efficiency and expanded accessibility in situations in which ethanol is more costly or otherwise not available.

**KEYWORDS**  16S rRNA gene, automation, contamination, large-scale studies, metagenomics, metabolomics, microbiome, sample storage, study design

Microbiome research continues to reveal important connections to human health and environmental sustainability (1–7). However, large-scale studies with high statistical power are needed to distinguish true associations from confounders (8–11). A key challenge is the dynamic nature of microbial communities, which vary across hosts, environments, and time points, a variability that clinical sampling alone cannot fully capture. Collecting samples outside clinical settings is therefore essential for population-based research, enabling broader participant diversity and more representative data sets. At-home collection broadens participant diversity, incorporating differences in ethnicity and age, among other factors. However, large-scale studies face challenges in sample acquisition, preservation, processing, and analysis (12–16).

We previously demonstrated that using 1 mL barcoded tubes significantly reduces well-to-well contamination between samples and enables the extraction of both nucleic acids and metabolites from a single tubed sample (16). Coined as the Matrix Method, this approach has shown to reduce processing time by half and integrate seamlessly with automated infrastructure (16). Small metabolite and microbial DNA extractions are often performed separately using biological replicates, making sample accessioning and metadata curation time-consuming and error-prone, and reducing correlation among data derived from each type of extraction. High-throughput integration of metabolomic and metagenomic data accelerates the discovery of molecular mechanisms underlying these associations (17). The previously benchmarked, Plate-based Method is time-consuming and labor-intensive, requiring manual transfer of swab heads into two separate 96-deep-well plates—one designated for metabolomics and the other for metagenomics (18, 19). Using the same reagents for nucleic acids extraction (i.e., from the Thermo MagMAX Microbiome Ultra 96-sample Kit, see Materials and Methods), our improved 1 mL Matrix Method eliminates this tedious sample plating step. It uses an automated opening and closing system of individual, barcoded, racked tubes known as Matrix tubes (Cat#: 3740, ThermoFisher Scientific, MA, USA). All 96 tubes in a rack can be scanned at once using a high-speed barcode reader (VisionMate, ThermoFisher Scientific, MA, USA). This provides a major speed advantage over the Plate-based Method, where each sample is plated and scanned individually. By eliminating the sample plating step of the Plate-based Method, errors introduced during this step are also eliminated. Furthermore, samples are fully contained within individual tubes, preventing any leakage of sample into neighboring samples during sample homogenization and cell lysis. Thus, the Matrix Method enables metabolomic and DNA extraction from a single swab, or 200 µL of sample, and is reliable, labor-saving, economical, and streamlined (16).

Our previous work highlights the benefits of the Matrix Method for microbiome sequencing in terms of reducing human error, processing time, and well-to-well contamination for human fecal samples (16). Similarly, we previously demonstrated the utility of the Matrix Method for sample storage and performing extractions for untargeted metabolomics. Using 95% ethanol instead of 50% methanol is safer (i.e., permits at-home sample collection) and reduces costs (i.e., can be stored and shipped at room temperature) (20). Here, we further extend the utility of the Matrix Method by exploring the potential benefits of incorporating different bead sizes during the cell lysis step to reduce taxon bias and by validating a third storage solution across a broader set of sample types, in addition to human feces.

To increase taxonomic recovery during extraction of nucleic acids, we included three different bead sizes during cell lysis (see Materials and Methods), which we anticipate will increase DNA yield from across a broader range of cell sizes. We prefer to use a mechanical approach instead of enzymatic lysis, because although enzymes, such as

chitinase, can increase recovery from pure cultures of certain fungi (21), enzymatic lysis has also been shown to reduce overall DNA yields (22) and is known to introduce taxon bias (23, 24). Bead-beating has a reduced cost and shorter processing time compared to enzymatic lysis (25, 26) and is the gold standard for taxonomically unbiased lysis of environmental samples (24).

To further reduce logistical challenges and the cost of shipping samples, a storage solution that stabilizes the microbial community at ambient temperature for an extended amount of time is essential (27, 28). Here, we compared sample storage in 95% (vol/vol) isopropanol to our previously benchmarked standard of 95% (vol/vol) ethanol at room temperature for one week. Ethanol has previously been shown to sufficiently preserve samples at room temperature for up to eight weeks (28) and serve as a suitable solvent for metabolite extractions (20). However, ethanol is sometimes not available and, in certain cases, has additional limitations due to use in beverages. In those cases, isopropanol can serve as a reliable substitute due to its more consistent availability, more stable and often lower cost, and reduced regulation compared to ethanol (29–32). To validate the modified cell lysis step for the Matrix Method and to investigate 95% isopropanol as a storage buffer for samples, we performed this study to directly compare the updated Matrix Method against the Plate-based Method. Specifically, we compared DNA yield, the number of quality sequences, microbial alpha- and beta-diversity, and microbial taxonomic composition. We tested each method against samples stored in either 95% ethanol or 95% isopropanol, across commonly collected sample types including built environment surfaces, human skin, human saliva, and feces from both mice and humans (Table 1). To extend our findings to untargeted metabolomics (i.e., LC-MS/MS), we also compared metabolite alpha- and beta-diversity and composition between 95% ethanol and 95% isopropanol for just the Matrix Method.

## RESULTS

In total, we processed fecal samples from four human and four mouse subjects; skin swabs from the armpits and hands of four human subjects; saliva samples collected from three human subjects both before and after tooth brushing; surface samples from four distinct areas within a research facility at University of California, San Diego; and swabs from four different keyboards that each belonged to a unique individual (Table 1).

We found DNA yield to be comparable across sample storage buffers (i.e., isopropanol and ethanol) and extraction protocols (i.e., Matrix Method and Plate-based Method) for all sample types except for skin swabs from human hands, for which storage in ethanol

**TABLE 1** Experimental design showing sample types, number of subjects, biological replicates, and negative controls per method and storage solution[a]

| Sample type | # of subjects | # of replicates per plate | # of extractions per method and storage solution | | | |
|---|---|---|---|---|---|---|
| | | | Plate-based Method metagenomics | | Matrix Method metabolomics and metagenomics | |
| | | | Ethanol | Isopropanol | Ethanol | Isopropanol |
| Human feces | 4 | 3 | 12 | 12 | 12 | 12 |
| Mouse feces | 4 | 3 | 12 | 12 | 12 | 12 |
| Saliva, before brushing | 3 | 3 | 9 | 9 | 9 | 9 |
| Saliva, after brushing | 3 | 3 | 9 | 9 | 9 | 9 |
| Skin, armpit | 4 | 3 | 12 | 12 | 12 | 12 |
| Skin, hand | 4 | 3 | 12 | 12 | 12 | 12 |
| Surface, floor | 4 | 3 | 12 | 12 | 12 | 12 |
| Surface, keyboard | 4 | 3 | 12 | 12 | 12 | 12 |
| Total samples | 30 | 24 | 90 | 90 | 90 | 90 |
| Negative control | NA[b] | NA | 12 | | 12 | |
| Total extractions | | | 192 | | 192 | |

[a]Negative controls consist of one blank well/tube per column of each method. All samples were kept in each storage solution at ambient temperature for eight days prior to processing. For the Plate-based Method, samples were collected on dual swabs (BD Falcon Swube Applicators) to accommodate plating into separate plates for metabolomics and metagenomics.
[b]NA, not applicable.

yielded slightly more DNA than storage in isopropanol for the Matrix Method (Wilcoxon signed-rank test, $W$ = 112.5, $P$ = 0.02) (Fig. 1). When comparing DNA yield between extraction protocols for replicates of the same samples that were stored in the same storage buffer, we found strong, positive correlations and no difference for samples stored in either ethanol or isopropanol (Fig. S1A). When considering DNA yield between sample storage buffers for replicates of the same sample that were extracted with the same method, we also found strong, positive correlations and no difference in yield for both the Matrix Method (Fig. S1B) and Plate-based Method (Fig. S1C). However, we note a slight trend for ethanol to recover more DNA when using the Plate-based Method (Fig. S1C).

We also found sequenced read counts to be strongly, positively correlated and comparable between replicate samples across sample storage buffers and extraction protocols for both 16S amplicon and shotgun metagenomic sequence data (Fig. S2). We note a marginally stronger correlation but greater difference in 16S read counts between replicates stored in ethanol and isopropanol when extracted using the Matrix Method vs the Plate-based Method (i.e., Spearman's $r$ = 0.87 vs 0.76; two-sided, paired $t$ = 0.77 vs 0.27) (Fig. S2C and E). The pattern in our shotgun metagenomic data were for a marginally weaker correlation and greater difference using the Matrix Method vs the Plate-based Method (Spearman's $r$ = 0.56 vs 0.66; two-sided, paired $t$ = 1.29 vs 1.09) (Fig. S2D and F).

When considering microbial community diversity, we found estimates of alpha-diversity (i.e., Faith's phylogenetic diversity [PD]) to be highly correlated between extraction methods regardless of the storage buffer used, for both 16S (Fig. 2A) and shotgun metagenomics data (Fig. 2B). We note a trend for the Matrix Method to capture a greater diversity of microbial taxa from low-biomass samples compared to the Plate-based Method, particularly in our 16S data (Fig. 2A and B). We also found alpha-diversity to

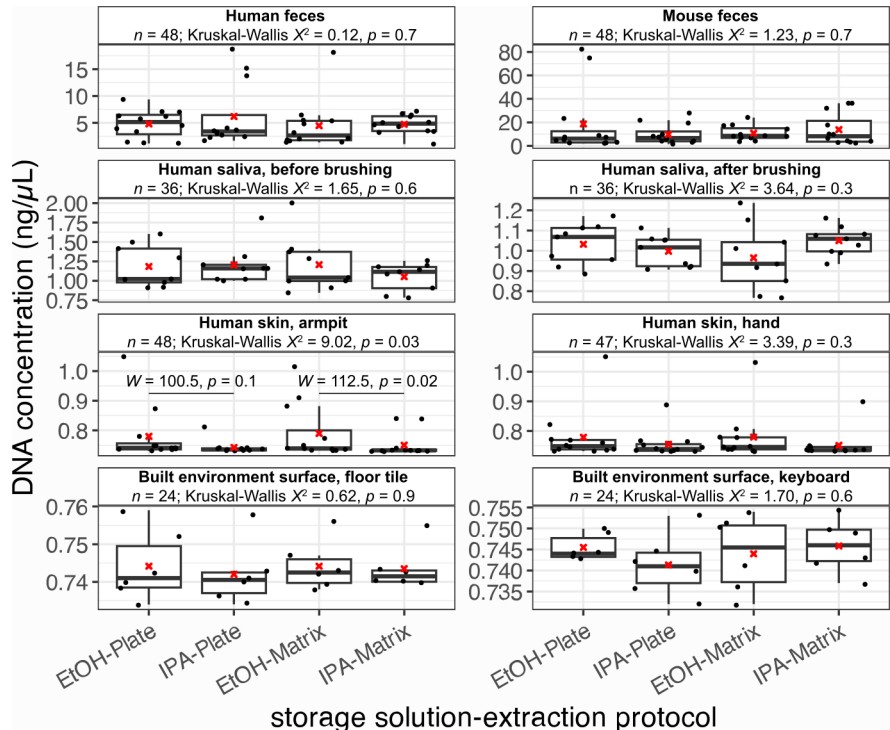

**FIG 1** DNA yield (ng/µL) per sample across sample types. Results from a Kruskal–Wallis test for differences in DNA yield among groups within each sample type are shown. For one sample type, the results from post-hoc Wilcoxon rank-sum tests comparing storage solutions within extraction protocols are shown. Red Xs indicate means. A miniaturized, high-throughput Quant-iT PicoGreen dsDNA assay was used for quantification, with a lower limit of 0.1 ng/µL; yields below this value were estimated by extrapolating from a standard curve. All samples were eluted into 70 µL of elution buffer.

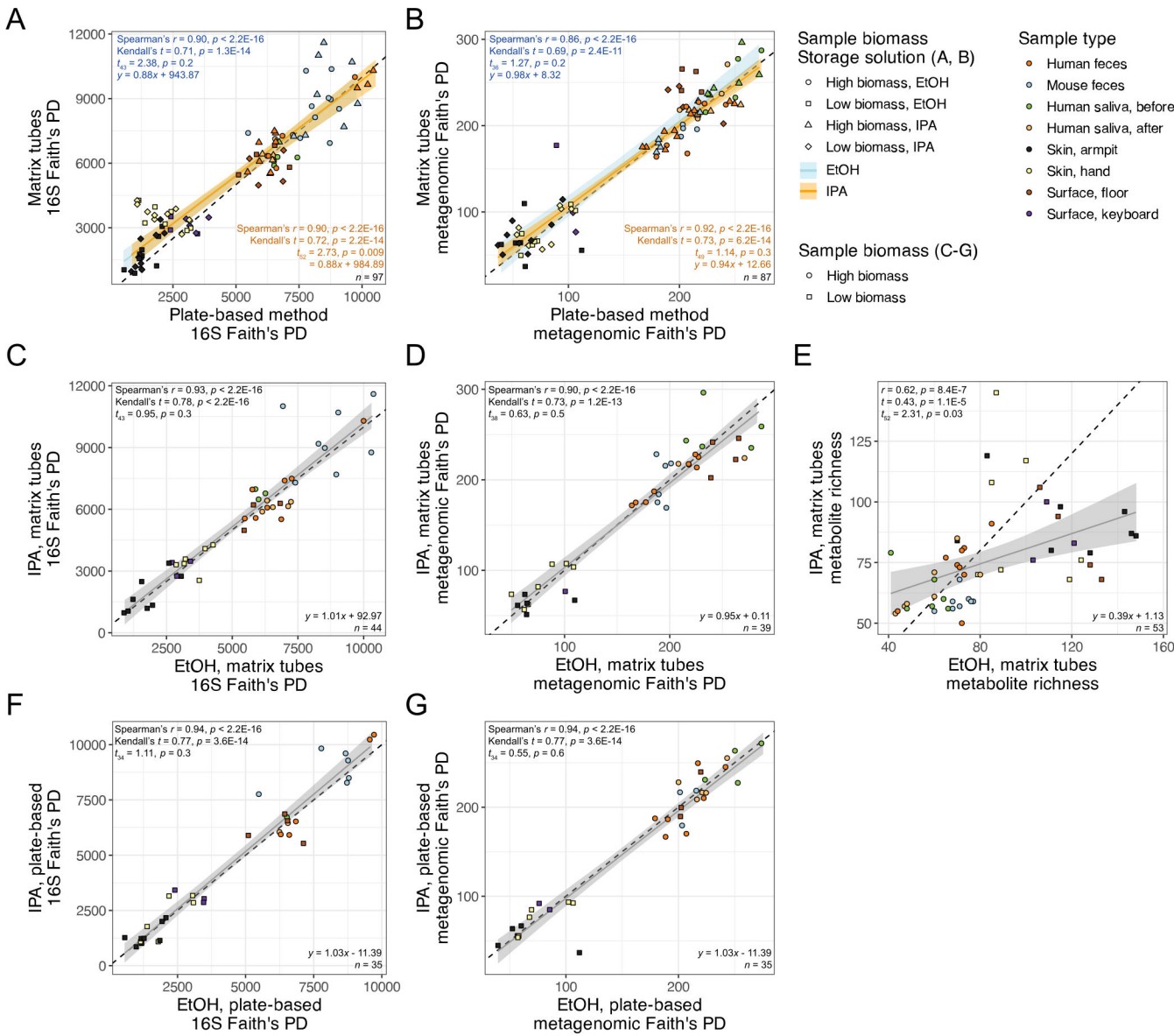

**FIG 2** Alpha-diversity comparisons between extraction protocols and storage solutions within each protocol for the same samples. The Matrix Method vs the Plate-based Method for (A) 16S data and (B) shotgun metagenomics data. Samples stored in ethanol (EtOH) vs isopropanol (IPA) extracted with the Matrix Method for (C) 16S data, (D) shotgun metagenomics data, and (E) untargeted LC-MS/MS metabolomics data. Samples stored in EtOH vs IPA extracted with the Plate-based Method for (F) 16S data and (G) shotgun metagenomics data. For each panel, results from correlation analyses and a two-sided, paired *t*-test are shown, as well as the equation for the linear model that best fits the data. Shaded areas are 95% confidence intervals for the linear models shown, and dotted lines represent $y = x$. PD, phylogenetic diversity. Sequence data were rarefied as in Table 2. Singletons were excluded from metabolomics data.

be highly consistent between replicate samples stored in different buffers but extracted with the same method for both 16S (Fig. 2C and F) and shotgun metagenomics data (Fig. 2D and G). Conversely, for LC-MS/MS metabolomics data, when comparing metabolite richness between replicates of the same sample stored in ethanol vs isopropanol for the Matrix Method, ethanol appears to capture a greater diversity of features from low-biomass samples compared to isopropanol (two-sided, paired $t = 2.31$, $P = 0.03$) (Fig. 2E). We reiterate that metabolomics was only performed on samples extracted using the Matrix Method; thus, we only present results regarding storage solutions across all sample types (see Materials and Methods).

To begin understanding the effects of sample storage buffer and extraction method on beta-diversity (i.e., sample-to-sample distances), we first examined how similarity among replicates of the same sample varied across methods. In general, for our microbial data sets, variation among replicates was often greater for low-biomass sample types and was more notable for all samples when using abundance-based distance metrics vs presence/absence-based ones (Fig. S3 and S4). For both 16S and shotgun metagenomics data, we observed differences across methods that were dependent on both sample type and distance metric (Fig. S3 and S4). For example, in our 16S data, we observed similar distances among skin samples from armpits extracted with the Plate-based Method and Matrix Method when using unweighted UniFrac distances and phylo-RPCA, but different values between methods when using Jaccard and weighted UniFrac distances (Fig. S3). In contrast, for our metabolomics data, we observed consistent differences in beta-diversity between storage solutions for skin samples from armpits and swabs from floor tiles, as well as with two of four distance metrics for swabs from computer keyboards (Fig. S5).

We then compared beta-diversity between replicates of the same sample subject to different storage solutions and extraction protocols using Mantel tests (Table 2). In general, Mantel correlations were strong between storage buffers and extraction methods across several distinct distance metrics (i.e., $r > 0.8$, $P = 0.0002$) and were stronger for high-biomass sample types compared to low-biomass ones (Table 2). For 16S data, for storage buffers and extraction methods, we observed a weaker correlation when using weighted UniFrac compared to other metrics for high-biomass sample types, as well as when using presence/absence-based distance metrics vs abundance-based metrics for low-biomass sample types (Table 2). For shotgun metagenomics data, we observed the same patterns for high-biomass sample types but found weaker correlations for abundance-based metrics vs presence/absence-based distance metrics for low-biomass sample types (Table 2). In contrast, we observed weak correlations between LC-MS/MS metabolomics profiles from different storage buffers extracted with the Matrix Method, particularly for low-biomass samples, despite being statistically significant (i.e., $P < 0.05$) for all distance metrics but Robust-Atchison PCA (RPCA) (Mantel $r = 0.08$, $P = 0.5$) (Table 2).

To further scrutinize how the use of different storage buffers and extraction methods affects microbial community composition, we assessed the effect of those factors on beta-diversity with permutational multivariate analysis of variance (PERMANOVA). For both 16S and shotgun metagenomics data, we observed a major difference in the effects of extraction method and host subject identity for low-biomass vs high-biomass sample types (Tables S1 and S2). For high-biomass sample types, host subject identity had the strongest effect and was the only factor found to be significant, except for oral swabs from humans. For just one of five distance metrics assessed (i.e., weighted UniFrac), our 16S data revealed that for oral swabs before brushing, the effect of sample storage buffer (pseudo-$F = 17.67$, $P = 0.002$) was nearly as strong as the host subject (pseudo-$F = 20.74$, $P = 0.001$). Although the effect was small, oral swabs after brushing were affected by extraction method (pseudo-$F = 5.83$, $P = 0.01$) (Table S1). Our shotgun metagenomics data also indicated that oral swabs before brushing were strongly affected by sample storage buffer for three of five metrics assessed (i.e., Jaccard, unweighted UniFrac, weighted UniFrac); however, oral swabs after brushing were not affected by extraction method but rather by storage buffer, again only for weighted UniFrac distances (Table S2).

For low-biomass sample types, we observed significant effects of sample storage buffer and extraction method dependent on the data type (i.e., 16S vs shotgun metagenomics), distance metric, and sample type (Tables S1 and S2). Our 16S data indicates that swabs of human skin (armpit and hand) are influenced by extraction method—with effects similar to host subject identity using presence/absence-based distance metrics, but weaker effects with abundance-based metrics (Table S1). Swabs of floor tiles were also affected by extraction method, which had similar effects as

**TABLE 2** Mantel correlations of sample-to-sample distances for extraction protocols (i.e., plate-based vs matrix) and storage solutions (i.e., ethanol vs isopropanol) for both 16S and shotgun metagenomic data[a]

| Data type | Comparison | Biomass | Metric | N | R | P-value |
|---|---|---|---|---|---|---|
| 16S | Extraction protocols | High | Jaccard | 44 | 0.97 | 0.0002 |
| | | | RPCA | 43 | 0.92 | 0.0002 |
| | | | UniFrac | 44 | 0.97 | 0.0002 |
| | | | Weighted UniFrac | 44 | 0.75 | 0.0002 |
| | | | Phylo-RPCA | 43 | 0.98 | 0.0002 |
| | | Low | Jaccard | 53 | 0.61 | 0.0002 |
| | | | RPCA | 53 | 0.89 | 0.0002 |
| | | | UniFrac | 53 | 0.57 | 0.0002 |
| | | | Weighted UniFrac | 53 | 0.80 | 0.0002 |
| | | | Phylo-RPCA | 53 | 0.80 | 0.0002 |
| | Storage solutions | High | Jaccard | 38 | 0.97 | 0.0002 |
| | | | RPCA | 36 | 0.92 | 0.0002 |
| | | | UniFrac | 38 | 0.98 | 0.0002 |
| | | | Weighted UniFrac | 38 | 0.75 | 0.0002 |
| | | | Phylo-RPCA | 36 | 0.98 | 0.0002 |
| | | Low | Jaccard | 41 | 0.75 | 0.0002 |
| | | | RPCA | 40 | 0.92 | 0.0002 |
| | | | UniFrac | 41 | 0.73 | 0.0002 |
| | | | Weighted UniFrac | 41 | 0.89 | 0.0002 |
| | | | phylo-RPCA | 40 | 0.91 | 0.0002 |
| Shotgun metagenomics | Extraction protocols | High | Jaccard | 46 | 0.98 | 0.0002 |
| | | | RPCA | 45 | 0.99 | 0.0002 |
| | | | UniFrac | 46 | 0.97 | 0.0002 |
| | | | Weighted UniFrac | 46 | 0.84 | 0.0002 |
| | | | Phylo-RPCA | 45 | 0.99 | 0.0002 |
| | | Low | Jaccard | 41 | 0.69 | 0.0002 |
| | | | RPCA | 41 | 0.55 | 0.0002 |
| | | | UniFrac | 41 | 0.67 | 0.0002 |
| | | | Weighted UniFrac | 41 | 0.22 | 0.0002 |
| | | | Phylo-RPCA | 41 | 0.64 | 0.0002 |
| | Storage solutions | High | Jaccard | 41 | 0.95 | 0.0002 |
| | | | RPCA | 40 | 0.96 | 0.0002 |
| | | | UniFrac | 41 | 0.94 | 0.0002 |
| | | | Weighted UniFrac | 41 | 0.77 | 0.0002 |
| | | | Phylo-RPCA | 40 | 0.96 | 0.0002 |
| | | Low | Jaccard | 33 | 0.83 | 0.0002 |
| | | | RPCA | 33 | 0.65 | 0.0002 |
| | | | UniFrac | 33 | 0.80 | 0.0002 |
| | | | Weighted UniFrac | 33 | 0.25 | 0.02 |
| | | | Phylo-RPCA | 33 | 0.69 | 0.0002 |
| LC-MS/MS metabolomics | Storage solutions | High | Jaccard | 42 | 0.43 | 0.0002 |
| | | | Bray-Curtis | 42 | 0.34 | 0.0002 |
| | | | RPCA | 42 | 0.38 | 0.0002 |
| | | | Cosine | 42 | 0.10 | 0.007 |
| | | | Canberra-Adkins | 42 | 0.44 | 0.0002 |
| | | Low | Jaccard | 34 | 0.24 | 0.01 |
| | | | Bray-Curtis | 34 | 0.21 | 0.004 |
| | | | RPCA | 34 | −0.08 | 0.5 |
| | | | Cosine | 34 | 0.12 | 0.02 |

*(Continued on next page)*

**TABLE 2** Mantel correlations of sample-to-sample distances for extraction protocols (i.e., plate-based vs matrix) and storage solutions (i.e., ethanol vs isopropanol) for both 16S and shotgun metagenomic data[a] (*Continued*)

| Data type | Comparison | Biomass | Metric | N | R | P-value |
|---|---|---|---|---|---|---|
| | | | Canberra-Adkins | 34 | 0.17 | 0.06 |

[a]For 16S data, high- and low-biomass samples were rarefied to 20,636 and 277 quality-filtered reads per sample (or had samples with fewer than 20,636 and 277 reads excluded, for RPCA and phylo-RPCA), respectively. For metagenomic data, high- and low-biomass samples were rarefied to 1,515,275 and 55,892 quality-filtered reads per sample (or had samples with fewer than 1,515,275 and 55,892 reads excluded, for RPCA and phylo-RPCA), respectively. For metabolomics data, singleton features were excluded. Rarefaction depths were selected to maintain at least 75% of samples for each sample type. Spearman's rank correlation coefficient is shown. Each *P*-value is from a permutation test with 50,000 iterations.

host subject with most distance metrics, but was the only significant factor with RPCA (Table S1). Swabs of keyboards were only influenced by extraction method (i.e., there was no effect of host subject identity), which had a small effect with presence/absence-based distance metrics, but a stronger effect with abundance-based metrics (Table S1). Our shotgun metagenomics data indicate that swabs of the armpit were influenced by extraction method about as much as host subject identity for all metrics except RPCA and phylo-RPCA, whereas for swabs of the hand, extraction method showed weaker effects than host subject, albeit significant for all distance metrics except UniFrac and phylo-RPCA (Table S2). Similar to our 16S data, shotgun metagenomics revealed significant and similar effects for extraction method and host subject identity for swabs of floor tiles, for all distance metrics but RPCA, the one distance metric for which extraction method was the only significant factor (Table S2). Shotgun metagenomics also revealed a weak but-significant effect of sample storage buffer for swabs of floor tiles when using weighted UniFrac distances (Table S2). Similarly, we found no effect of host subject identity and a significant interaction between sample storage buffer and extraction method for swabs of keyboards, for all distance metrics except weighted UniFrac (Table S2).

When assessing the impact of sample storage buffer on LC-MS/MS metabolite composition using PERMANOVA, we found that only host subject identity was a significant factor for high-biomass sample types, and its effect varied among specific sample types and distance metrics (Table S3). For low-biomass sample types, the effect of host subject identity was often lacking and only significant for swabs from armpits for certain distance metrics (Table S2). However, the effect of sample storage buffer was significant for swabs of the armpit, swab of the hand when using Jaccard distances, and swabs from surfaces (Table S2).

To further explore bias in the recovery of specific microbes and metabolites from samples processed with different storage solutions and extraction methods, we performed analyses to reveal which features were exclusive to any one method. For microbes, we used our shotgun metagenomics data to explore taxon bias for Archaea, Bacteria, and Fungi (Fig. 3 to 7). When visualizing these patterns across the Web of Life phylogeny for Archaea and Bacteria, we found that although the majority of species were recovered across all methods, many were exclusive to a single method, with most exclusive microbes recovered from samples stored in ethanol and extracted with the Matrix Method (Fig. 3). For example, although the Actinomycetota are generally recovered well from all protocols, early-diverging clades in the Bacteroidota, as well as several scattered within the Pseudomonadota, show strong affinity for only one method, with the majority of taxa recovered from samples stored in ethanol and extracted using the Matrix Method (Fig. 3). When we quantified these differences, we found that, in addition to recovering the most species overall across all methods, storage in ethanol followed by extraction with the Matrix Method recovered the most unique species from all sample types examined, except those from the human oral cavity (Fig. 4 and 5). For the latter, storage in isopropanol recovered the most unique species, with subsequent extraction using the Plate-based Method and Matrix Method recovering the most unique taxa from samples collected from before and after brushing, respectively (Fig. 4).

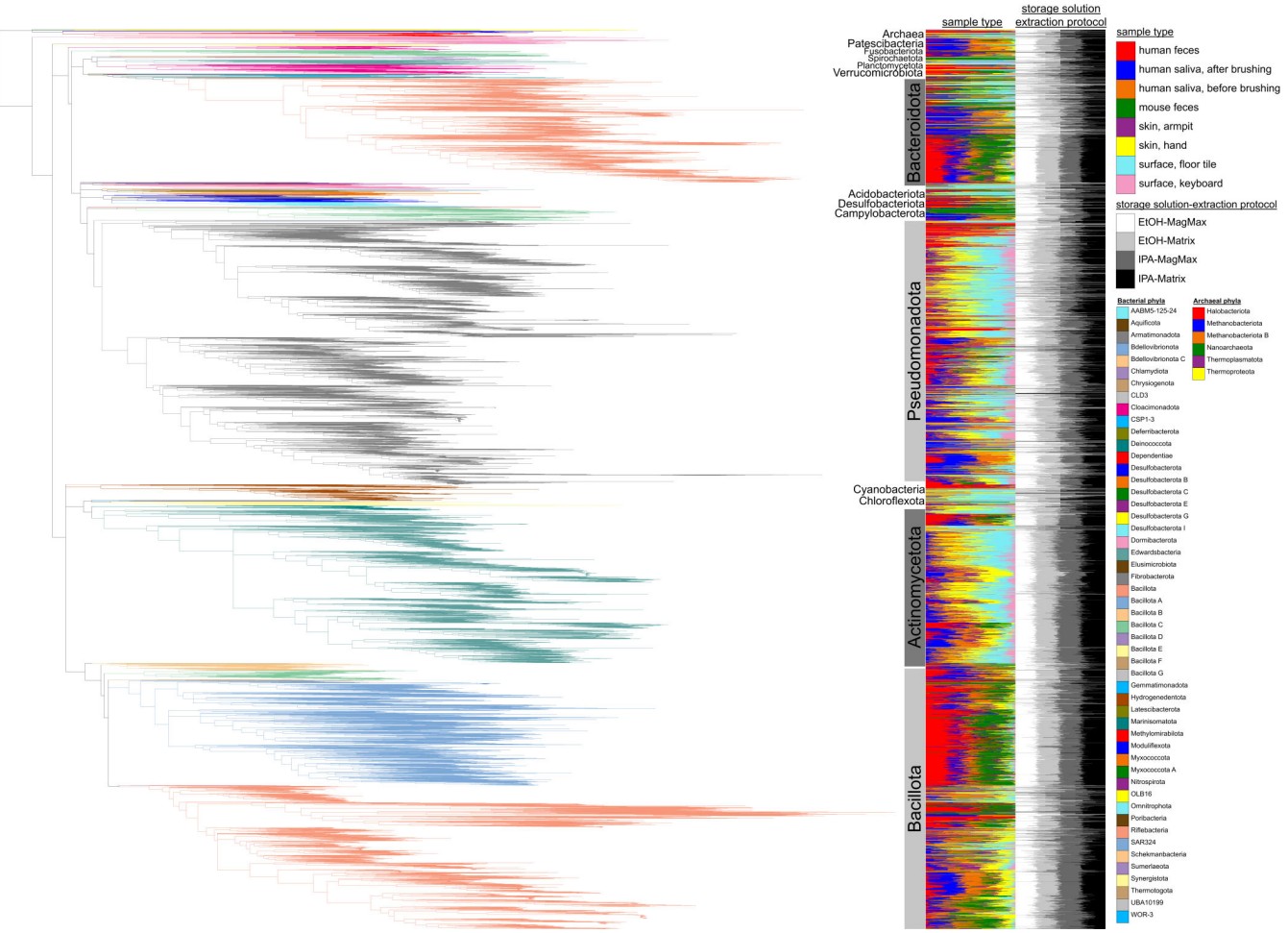

**FIG 3** Phylogenetic bias among storage solutions and extraction protocols, based on shotgun metagenomics data. The microbial phylogeny includes Archaea and Bacteria, with colors highlighting distinct phyla. Each tip in the phylogeny represents a unique microbial strain. For each strain, we indicate phylum, one bar graph indicating which sample types the strain was represented in, and a second bar graph indicating which storage solution-extraction protocol pair the strain was represented in.

For fungi, storage in ethanol followed by extraction with the Matrix Method recovered the most genera from all sample types examined except for human feces, for which storage in isopropanol vs ethanol recovered 13 vs 12 fungal genera, and human saliva before brushing, for which storage in isopropanol followed by extraction with the Plate-based Method recovered 13 genera compared to 8 or nine genera from other methods (Fig. 6 and 7). Examination of each subset of genera indicates that each method recovers a similar but unique assemblage of fungi (Fig. S6). Similarly, for the number of exclusive fungal genera recovered, storage in ethanol followed by extraction with the Matrix Method recovered the greatest number of fungal genera from all sample types except for human feces, for which storage in isopropanol recovered 13 vs 12 genera when stored in ethanol, and human saliva before brushing, where storage in either solution followed by extraction with the Plate-based Method recovered three exclusive fungal genera vs one when using the Matrix Method (Fig. 6 and 7). We note that although, for the majority of sample types, either extraction method recovered at least one exclusive fungal genus, the Matrix Method was unique in this regard for human feces and saliva after brushing (Fig. 6).

To identify which fungal genera showed variable recovery across methods, and in absence of a comprehensive phylogeny for fungi, we examined a heatmap of the abundance table used in our taxon bias analysis (Fig. S6). The most cosmopolitan taxa

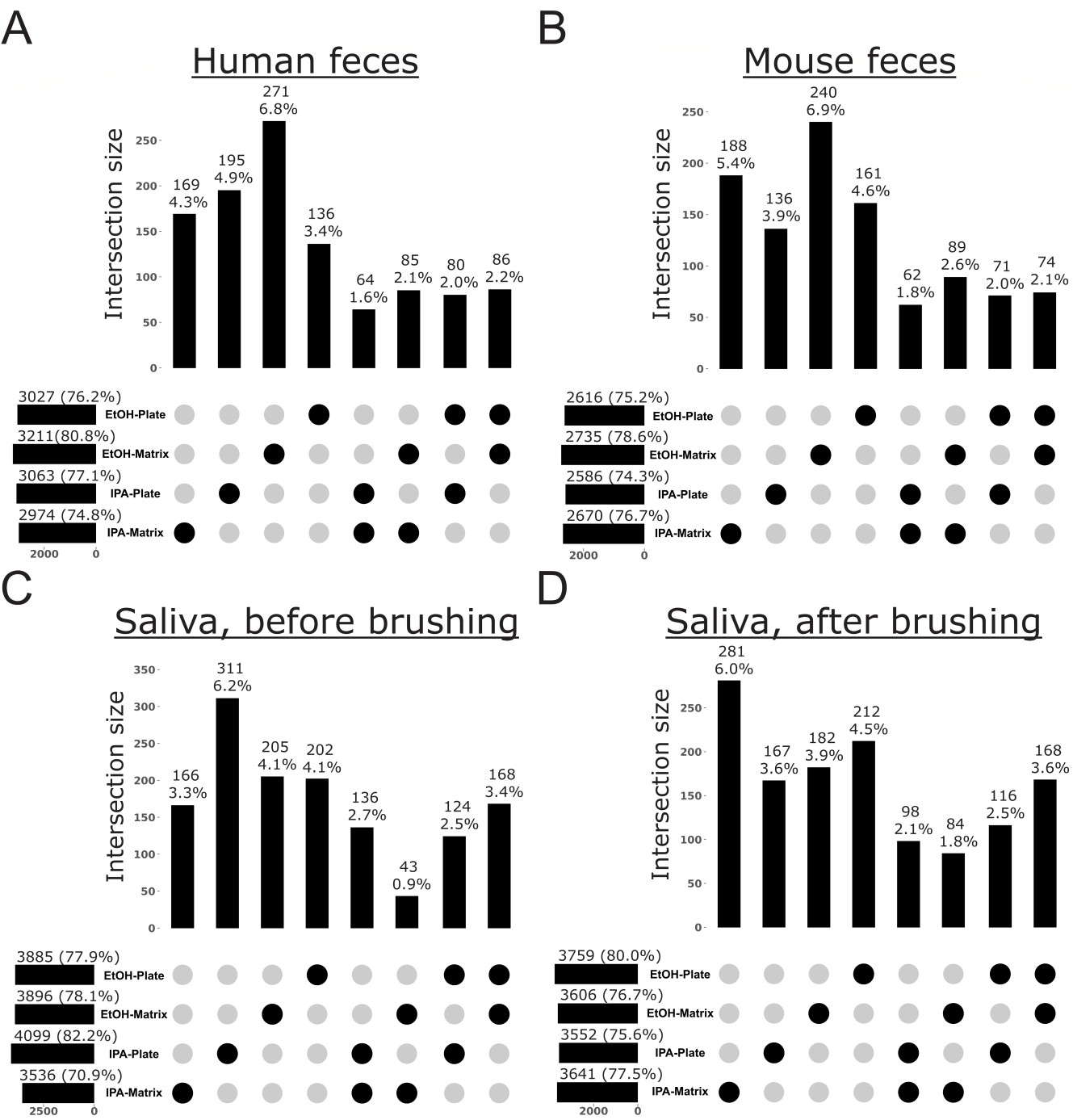

**FIG 4** Taxonomic bias among storage solutions and extraction protocols for high-biomass sample types, based on shotgun metagenomics data for Archaea and Bacteria. UpSet plots highlight the number of taxa exclusive to a single method or an intersection of methods, out of the total number of taxa recovered across all methods. For each method, horizontal bars indicate the total number of taxa recovered (i.e., taxon richness) and the percentage this represents of the total taxa recovered across all methods. Vertical bars indicate the number of taxa exclusive to a given method or intersection of methods, and the percentage this represents of the total taxa recovered across all methods. (A) Human feces. (B) Mouse feces. (C) Saliva before brushing teeth. (D) Saliva after brushing teeth.

were *Malassezia* and *Penicillium*, which were recovered from all samples except for mouse feces stored in isopropanol and extracted with the Plate-based Method, or mouse feces stored in ethanol and extracted with the Matrix Method, respectively (Fig. S6). Similarly, *Aspergillus* was recovered from all samples except for mouse feces stored in ethanol and extracted with the Plate-based Method, and human feces stored in either solution and extracted with the Plate-based Method (Fig. S6). We also explored which

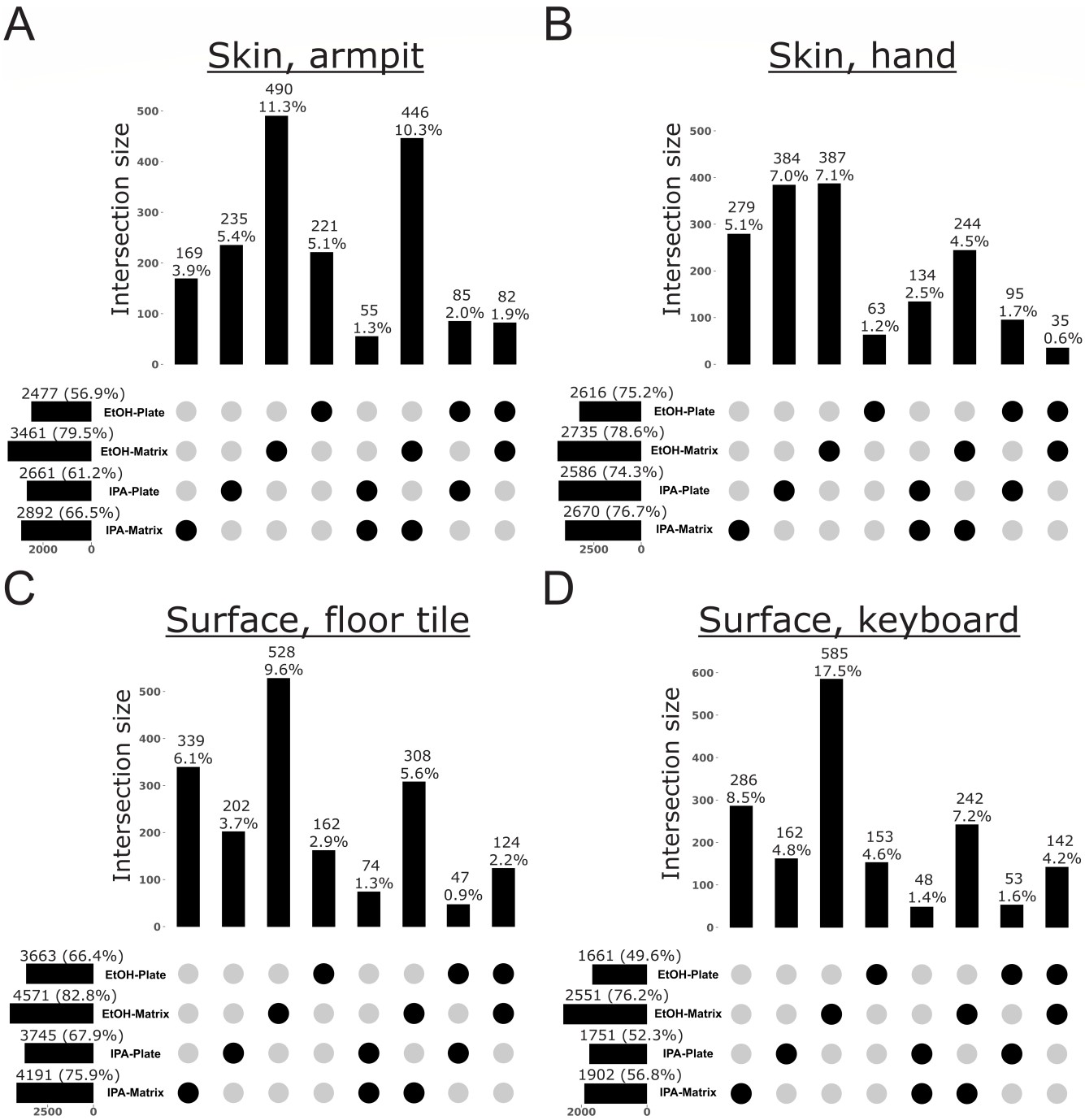

**FIG 5** Taxonomic bias among storage solutions and extraction protocols for low-biomass sample types, based on shotgun metagenomics data for Archaea and Bacteria. UpSet plots highlight the number of taxa exclusive to a single method or an intersection of methods, out of the total number of taxa recovered across all methods. For each method, horizontal bars indicate the total number of taxa recovered (i.e., taxon richness) and the percentage this represents of the total taxa recovered across all methods. Vertical bars indicate the number of taxa exclusive to a given method or intersection of methods, and the percentage this represents of the total taxa recovered across all methods. (A) Human skin, armpit. (B) Human skin, hand. (C) Surface swab, floor tile. (D) Surface swab, computer keyboard.

fungal genera were recovered from only one extraction method (i.e., exclusive genera) (Fig. 6 and 7; Fig. S6). A few fungi were recovered only when using the Plate-based Method, including *Alternaria* and *Exophiala* from human saliva before brushing and *Hirsutella* from mouse feces (Fig. S6). More frequently, certain fungal genera were recovered only when using the Matrix Method, including *Aspergillus* and *Botrytis* from

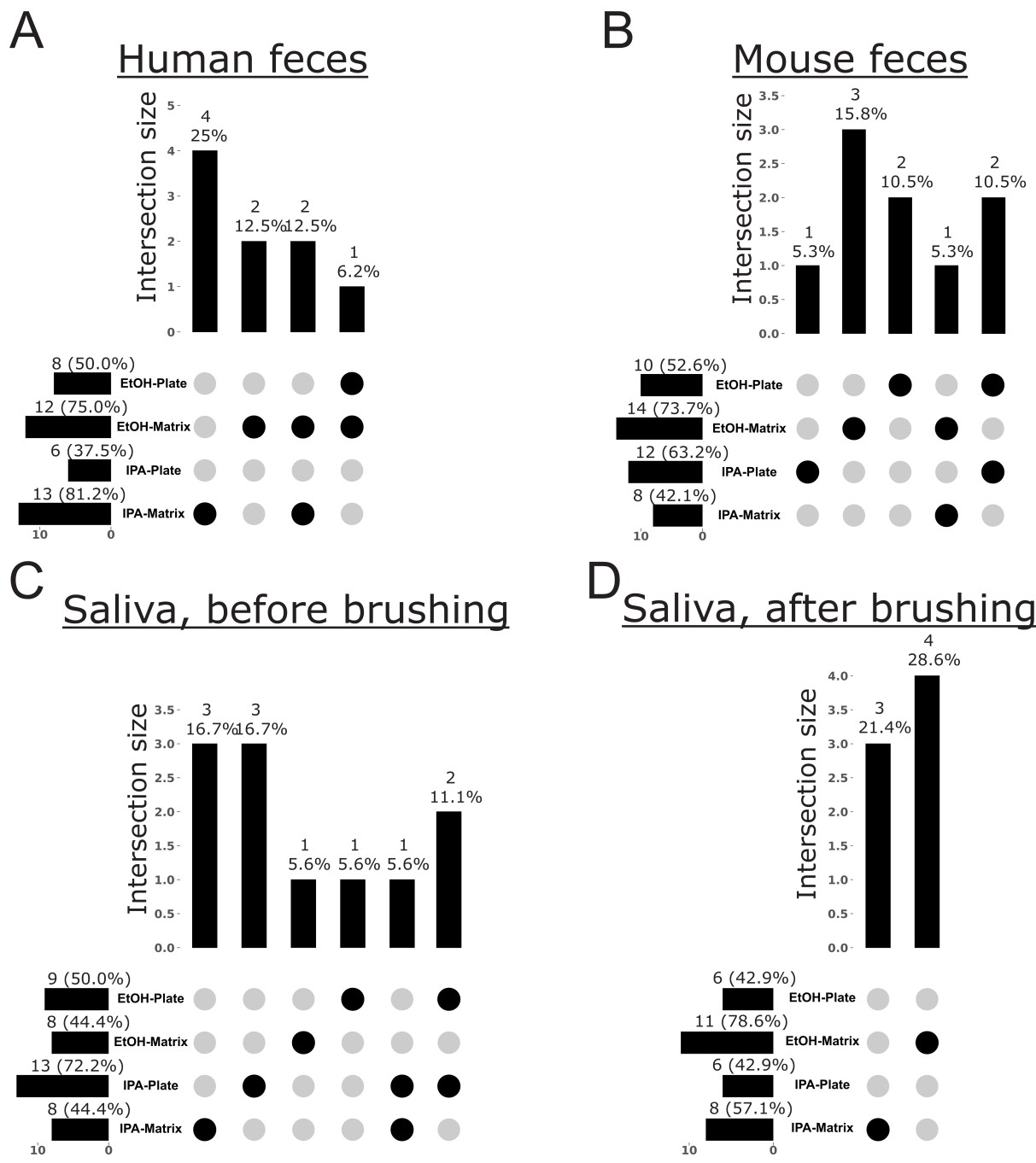

**FIG 6** Taxonomic bias among storage solutions and extraction protocols for high-biomass sample types, based on shotgun metagenomics data for fungi. UpSet plots highlight the number of taxa exclusive to a single method or an intersection of methods, out of the total taxa recovered across all methods. For each method, horizontal bars indicate the total number of taxa recovered (i.e., taxon richness) and the percentage this represents of the total taxa recovered across all methods. Vertical bars indicate the number of taxa exclusive to a given method or intersection of methods, and the percentage this represents of the total taxa recovered across all methods. (A) Human feces. (B) Mouse feces. (C) Saliva before brushing teeth. (D) Saliva after brushing teeth.

human feces; *Cercospora* from mouse feces; *Cercospora, Chaetomium, Diaporthe, Filobasidium, Lasodiplodia, Metarhizium, Moesziomyces, Pestalotiopsis, Pleurotus, Saccharomyces, Talaromyces, Trametes, Ustilaginoidea,* and *Wallemia* from armpit skin; *Cyberlindnera, Kwoniella,* and *Pochonia* from hand skin; *Amorphotheca, Bacidia, Cyphellophora, Emericellopsis, Hirsutella, Hyaloscypha, Kockovaella, Mycena, Neurospora, Parathielavia, Phialemonium, Pochonia, Scedosporium,* and *Trichosporon* from swabs of floor tiles; and *Ascochyta, Bipolaris, Boererria, Cercospora, Diaporthe, Exserohilum, Pleurotus, Trichosporon,*

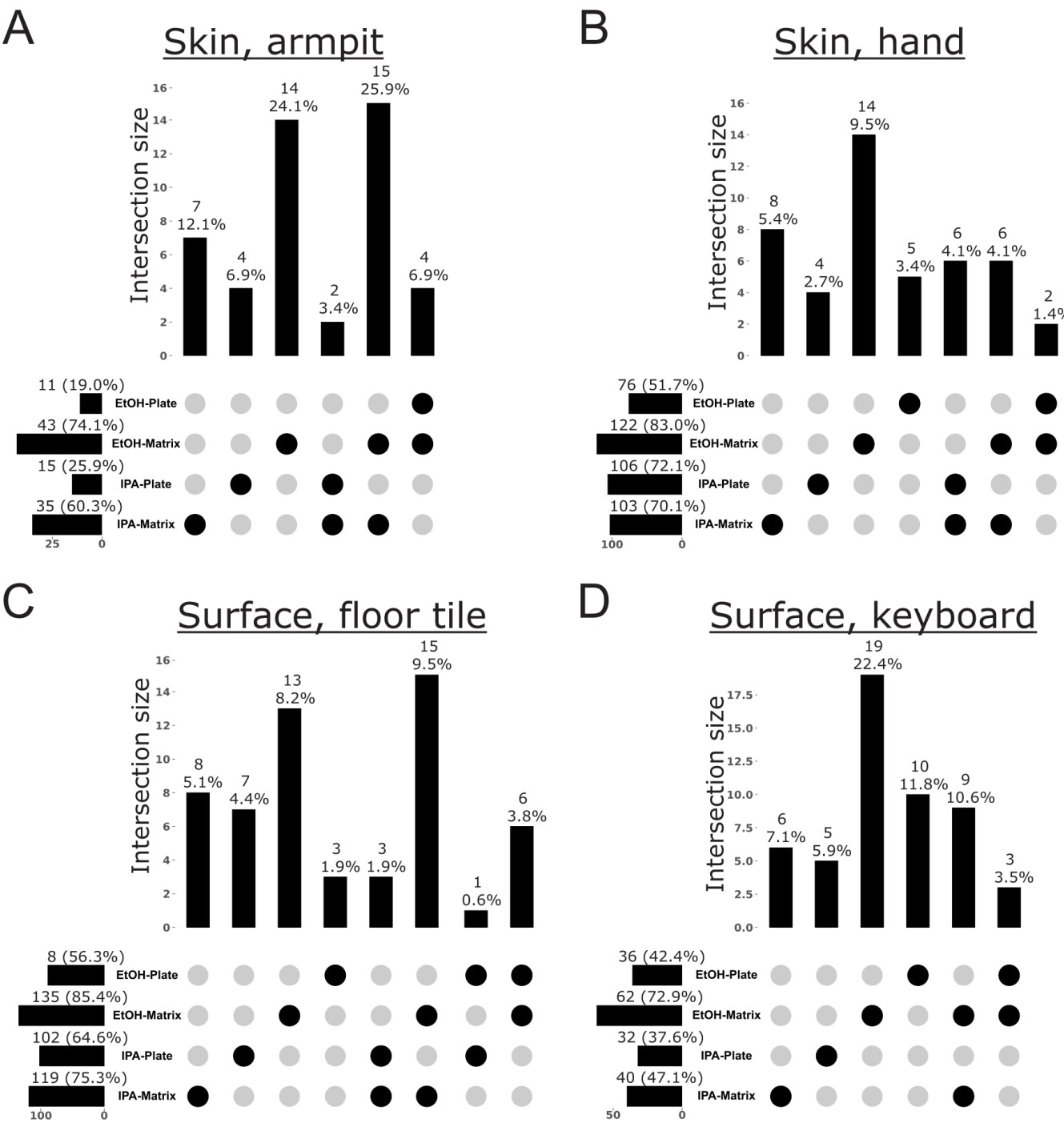

**FIG 7** Taxonomic bias among storage solutions and extraction protocols for low-biomass sample types, based on shotgun metagenomics data for fungi. UpSet plots highlight the number of taxa exclusive to a single method or an intersection of methods, out of the total taxa recovered across all methods. For each method, horizontal bars indicate the total number of taxa recovered (i.e., taxon richness), and the percentage this represents of the total taxa recovered across all methods. Vertical bars indicate the number of taxa exclusive to a given method or intersection of methods, and the percentage this represents of the total taxa recovered across all methods. (A) Human skin, armpit. (B) Human skin, hand. (C) Surface swab, floor tile. (D) Surface swab, computer keyboard.

and *Ustilaginoidea* from swabs of computer keyboards. There were also several genera that were recovered with just one storage solution-extraction method pair (Fig. 6 and 7; Fig. S6). Assessing exclusivity independently for each sample type and then tallying across all types, the majority of these exclusive fungal genera were recovered from samples stored in ethanol ($n = 101$) as opposed to isopropanol ($n = 73$), and that were

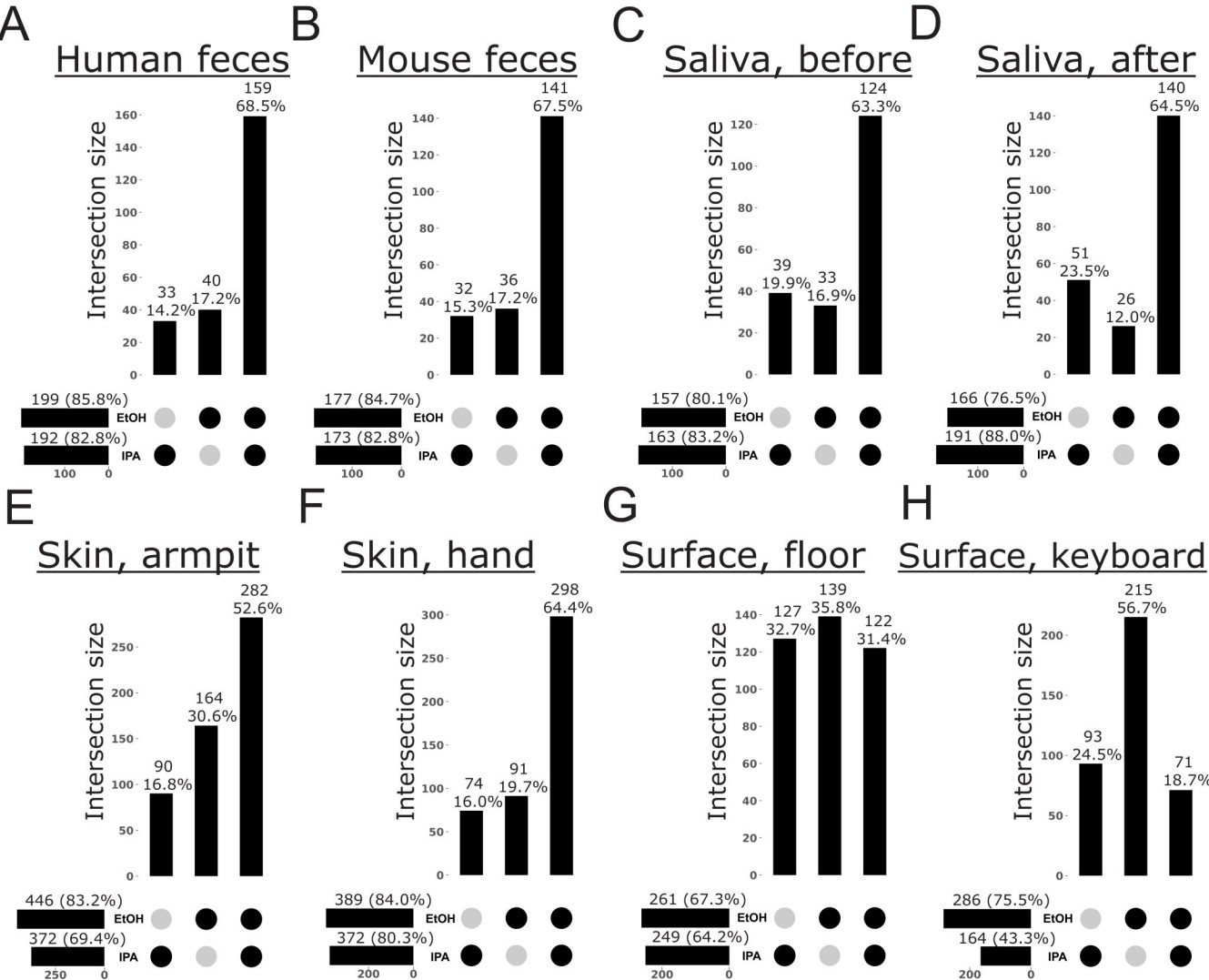

**FIG 8** Feature bias among storage solutions based on untargeted LC-MS/MS metabolomics data. UpSet plots highlight the number of metabolites exclusive to a single method or an intersection of methods, out of the total metabolites recovered across all methods. For each method, horizontal bars indicate the total number of metabolites recovered (i.e., metabolite richness) and the percentage this represents of the total metabolites recovered across all methods. Vertical bars indicate the number of metabolites exclusive to a given method or intersection of methods, and the percentage this represents of the total metabolites recovered across all methods. (A) Human feces. (B) Mouse feces. (C) Saliva before brushing teeth. (D) Saliva after brushing teeth. (E) Human skin, armpit. (F) Human skin, hand. (G) Surface swab, floor tile. (H) Surface swab, computer keyboard.

extracted using the Matrix Method (*n* = 119) as opposed to the Plate-based Method (*n* = 55) (Fig. 6 and 7; Fig. S6). Specifically, storage in ethanol followed by extraction with the Matrix Method recovered 73 exclusive fungal genera, whereas storage in isopropanol followed by extraction with the Matrix Method recovered just 28 exclusive genera (Fig. S6). For example, *Alternaria*, *Beauveria*, *Cercospora*, and *Exserohililum* were only recovered from human feces when stored in isopropanol and extracted with the Matrix Method, whereas *Acaromyces* and *Colletotrichum* were only recovered when stored in ethanol and extracted with the Matrix Method (Fig. S6). Similarly, storage in ethanol followed by extraction with the Plate-based Method recovered 46 exclusive fungal genera, whereas storage in isopropanol followed by extraction with the Plate-based Method recovered 27 exclusive genera (Fig. S6). For example, *Filobasidium* and *Fusarium* were only detected

from human saliva before brushing with storage in ethanol and extraction with the Plate-based Method, whereas *Agaricus* and *Leptographium* were only recovered with storage in isopropanol and extraction with the Plate-based Method (Fig. S6).

We observed a similar pattern of recovery bias for metabolites with respect to storage solutions, where ethanol recovered more total metabolites than isopropanol and more exclusive metabolites than isopropanol, from all sample types examined, except those from the human oral cavity (Fig. 8). For the surfaces of computer keyboards, storage in isopropanol recovered only half as many metabolites compared to storage in ethanol (Fig. 8). For human saliva after brushing, skin of the armpit, and the surfaces of computer keyboards, the difference in the number of exclusive metabolites between storage solutions was nearly twofold (Fig. 8). We note that for surface samples only, each storage solution recovered a greater number of exclusive metabolites than were recovered by both solutions (Fig. 8). Considering the identity of metabolites, the recovery of a greater number of metabolites after storage in ethanol vs isopropanol was more drastic for low-biomass sample types (Fig. S7). For example, the subclasses "Amino acids, peptides, and analogues," "Aryl bromides," "Ceramides," and "Tetraterpenoids" were enriched following storage in ethanol for all low-biomass sample types (Fig. S7). Other subclasses with variable recovery between sample storage solutions included "Benzoic acids and derivatives," "Bile acids, alcohols, and derivatives," and "Fatty acids and conjugates" for human feces; "Fatty alcohols" and "Hydroxycinnamic acids and derivatives" for mouse feces; "Carbohydrates and carbohydrate conjugates" for human saliva before brushing; and "Benzoic acids and derivatives" and "Fatty alcohols" for human saliva after brushing (Fig. S7).

## DISCUSSION

In this study, we benchmarked the Matrix Method against the previously established Plate-based Method for DNA and metabolite extraction across diverse microbiome-relevant sample types. Our findings support previous findings that the Matrix Method is a robust, scalable, and cost-effective alternative for high-throughput microbiome studies (16). The key advantage to the Matrix Method lies in its compatibility with automation, reduced risk of contamination, and streamlining of both metagenomic and metabolomic workflows from a single sample tube.

In terms of DNA yield, we observed minor trends for the Plate-based Method to yield more DNA than the Matrix Method, and for ethanol to outperform isopropanol within each extraction method (Fig. S1). The former observation may be attributed to differences in swabs between methods, as the swabs used for the Plate-based Method have a larger sampling surface. Still, sequenced read counts from samples stored in different storage solutions and/or extracted with different methods were highly correlated (Fig. S2). Correlations between methods were slightly stronger for 16S data compared to shotgun metagenomics data, likely due to the higher taxonomic resolution inherent in the latter. Similarly, results from community diversity analyses indicate that microbiome data from each storage solution and/or extraction method are highly concordant (e.g., Fig. 2; Fig. S2; Table 2). However, for all sample types except those from the human oral cavity, storage in ethanol followed by extraction using the Matrix Method results in the highest alpha-diversity and highest proportion of unique taxa (Fig. 2 to 5). The better performance of the Matrix Method for low-biomass sample types in this regard is especially apparent (Fig. 2 and 5). We suspect that the increased recovery of taxa when using the Matrix Method is due to the modified lysis step introduced to that protocol in this study, which we expected would lyse a broader diversity of microbes.

Microbial community composition was highly consistent across both extraction protocols and storage solutions for 16S and shotgun metagenomic data (Table 2), suggesting that neither protocol introduced significant bias in taxonomic composition for Archaea and Bacteria, which we discuss further below. However, all methods examining beta-diversity show weaker correspondence between replicate samples for low-biomass sample types (Fig. S3 and S4; Table 2; Tables S1 and S2), reflecting

the inherent challenges in collecting and processing such samples (12, 14, 15, 33). Interestingly, Mantel correlations for our 16S data were stronger for low-biomass sample types when using abundance-based distance metrics, whereas those for our shotgun metagenomics data for Archaea and Bacteria were stronger when using presence/absence-based metrics (Table 2). This discrepancy could be due to differences in taxonomic resolution or variation in the reliability of using read counts as proxies for abundance, depending on the data type. Previous studies benchmarking extraction methods have also found that, although one method generally outperforms the others, low-biomass sample types often exhibit context-dependent variation (18, 34).

Results of PERMANOVA best highlight the effects of different storage solutions and extraction methods for low-biomass sample types (Tables S1 and S2). For microbes, human oral samples appear to be influenced by sample storage buffer and/or extraction method, although results for samples taken after brushing were less clear (Tables S1 and S2), perhaps due to lower microbial load following oral hygiene activities, which appears to be supported by our analysis of alpha-diversity (i.e., the total number of archaeal + bacterial taxa is greater before vs after brushing), except for when samples were stored in isopropanol and extracted with the Matrix Method (Fig. 4). Notably, for keyboard surfaces, extraction protocol emerged as a dominant factor influencing beta-diversity, underscoring the sensitivity of low-biomass environments to methodological variability. We hypothesize that the high number of replicate swabs made it difficult to maintain consistency across samples, particularly for keyboards, where each key represents a small surface area with potentially distinct and potentially limited microbial loads depending on frequency and pattern of use. This may be especially apparent for the Matrix Method, which has been previously shown to reduce sample-to-sample contamination that may otherwise homogenize samples with different microbial community composition, particularly for communities with relatively low alpha-diversity (16). As support for this, although not consistent across all distance metrics, we observed greater beta-diversity among low-biomass samples extracted using the Matrix Method in our 16S data from surface swabs of keyboards (using weighted UniFrac distances; Fig. S3), and in our shotgun metagenomic data from swabs of human skin for the majority of distance metrics examined (Fig. S4).

Our analyses of taxon bias support storage in ethanol followed by extraction with the Matrix Method as optimal for the recovery of Archaea, Bacteria, and Fungi from all sample types, except from the human oral cavity (Fig. 4 to 7). For those sample types, a greater number of archaeal species, bacterial species, and fungal genera were recovered from saliva before brushing when stored in isopropanol and extracted using the Plate-based Method (Fig. 4 and 6). Additionally, a greater number of archaeal and bacterial species were recovered from saliva after brushing when stored in isopropanol and extracted using the Matrix Method (Fig. 4). We note that for cases in which variable recovery of certain taxa does not align with one particular method, and in particular for low-biomass samples, comparisons of recovery for especially lowly abundant taxa may be confounded by low microbial load and limited sample sizes (Fig. 3; Fig. S6. For example, the absence of *Candida* in human saliva before brushing may be confounded by one of these factors, as it was only missing from samples stored in isopropanol and extracted using the Matrix Method (Fig. S6).

Although indirect, the results of our analyses of taxon bias support the use of a mixture of three different bead sizes (0.1, 0.5, and 1 mm) during sample lysis for nucleic acid extraction inherent to the Matrix Method (16). This approach may be more effective at lysing an assemblage of microbial propagules spanning a broad range of morphologies and sizes compared to the single size (i.e., 0.1 mm) used in the Plate-based Method. To support this, we found that for mouse feces, human saliva after brushing, armpit skin, swabs from floor tiles, and the surfaces of computer keyboards (i.e., 5/8 or 63% of sample types), using the Matrix Method recovered a greater number of archaeal and bacterial species, regardless of the storage solution used (Fig. 4 and 5). Similarly, for human feces, saliva after brushing, armpit skin, swabs of floor tiles, and the surfaces of computer

keyboards (i.e., 5/8 or 63% of sample types), using the Matrix Method recovered a greater number of fungal genera regardless of the storage solution (Fig. 6 and 7). Further and across all sample types, a greater number of exclusive fungal genera were extracted using the Matrix Method ($n = 119$) as opposed to the Plate-based Method ($n = 55$) (Fig. S6). Given these findings, we suggest that future studies including multiple sample types, or those aiming to perform meta-analyses with other data sets, use ethanol as a storage solution followed by either extraction method, with preference given to the Matrix Method if workflows allow (see Materials and Methods and 16). Similarly, future microbiome studies focused on a single sample type included here should consider our taxon bias results reported separately for archaeal + bacterial species and fungal genera and select the most optimal approach (Fig. 4 to 7). For example, we observed measurable differences in the recovery of important taxa, such as certain members of the Pseudomonadota (Fig. 3), as well as *Alternaria*, *Penicillium*, and *Saccharomyces* (Fig. S6).

For metabolites, isopropanol was drastically inferior to ethanol as a storage solution in terms of alpha-diversity and taxon bias, except for samples from the human oral cavity (Fig. 2 and 8; Fig. S7). Although previous work has shown storage in ethanol to perform similarly to methanol, the commonly used solvent for extraction in metabolomics, the authors did not explicitly examine per-sample alpha-diversity (20). Instead, Zuffa et al. (20) found 95% ethanol and 50% methanol to recover a similar number of taxa across all fecal samples examined. Here, we used the same type of analysis to show that 95% ethanol recovers a greater number of unique metabolites, as well as a greater number of metabolites, compared to 95% isopropanol across all sample types except those from the human oral cavity (Fig. 8). This pattern was also consistent for metabolite subclasses (Fig. S7). We note that although we were not able to compare the Plate-based Method with the Matrix Method for metabolomics here, the previous study by Zuffa et al. (20) used the Matrix Method for benchmarking extraction with ethanol vs methanol, the latter performed using a plate-based approach. Furthermore, our observation that the wooden swabs used in the Plate-based Method contribute undesirable compounds to and absorb non-negligible amounts of storage solution leads us to conclude that the Matrix Method is suitable for metabolomics extractions using ethanol or methanol.

In contrast to our microbiome results, whereas metabolite composition was consistent between storage solutions for high-biomass sample types, we observed significant variation between samples stored in ethanol and isopropanol for low-biomass sample types (Fig. 3; Table 2; Table S3). This was expected given the differential solubility and stability of various metabolites in ethanol and isopropanol (35) and can be explained by our analysis of metabolite bias (Fig. 6). For high-biomass samples, ethanol performed slightly better in terms of metabolite recovery and overlap across replicates (Fig. S5 and S6). This reinforces prior findings that ethanol is an effective solvent for metabolomics (20), while also highlighting that isopropanol should only be used for storage of certain sample types. Although isopropanol may be more readily available in certain remote areas, our results should be considered when designing studies that rely on consistent metabolomic signatures. In addition to similar studies such as Zuffa et al. (20), these findings should also be considered alongside previously established and/or standardized protocols, such as those developed by the Earth Microbiome Project (31).

Together, these results validate the Matrix Method as a powerful alternative to traditional plate-based workflows. By enabling dual-omics extraction from a single sample, minimizing labor, and maintaining data quality across diverse conditions, the Matrix Method addresses several bottlenecks in microbiome research. Its compatibility with ambient storage and automation positions it well for large-scale, decentralized studies aimed at understanding microbiome function and dynamics across diverse populations and environments. In addition to the advantages identified in this study, we previously demonstrated how the Matrix Method can drastically reduce sample-to-sample contamination compared to the Plate-based Method (16).

There are also some limitations to this new approach that merit further discussion. First, the number of biological samples per sample type included in our study is limited

(Table 1). This is inherent in our approach for benchmarking new extraction methods, which limits the number of high- and low-biomass samples to a single 96-well plate (i.e., or one rack of matrix tubes) in order to avoid assessing a plate effect (20). As with many studies, increasing the number of samples per sample type may impact the results. Similarly, additional sample types should be explored, such as those from other animals, plants, and soil. Future studies should explore the costs/benefits of the Matrix Method with larger sets of samples per sample type, as well as sample types beyond those included here, in order to address this limitation.

Furthermore, although the Matrix Method is well adapted for automated, high-throughput sample processing, it is inherently an extraction method that uses single tubes instead of 96-well plates, as previously compared in other studies (12, 36). For example, Marotz et al. (36) compared a single-tube-based extraction method to a plate-based one and found that both methods generated comparable DNA yields, sequenced read counts, and microbial beta-diversity. However, the single-tube-based method required $>2\times$ the processing time compared to the plate-based method (36). Similarly, Minich et al. (12) compared single-tube-based extractions with plate-based methods in a previous study focused on sample-to-sample contamination. They reported that the single-tube method resulted in a drastically reduced number of events compared to Plate-based Methods (12). With the exception of the specific extraction kits used, the Plate-based Methods used in those other studies are similar to the Plate-based Method used here (12, 36). However, the single-tube-based methods used in those studies were manually processed using standard microcentrifuge tubes (e.g., 1.5 mL) (12, 36), rather than the type of smaller, racked matrix tubes used in the Matrix Method (16). In this regard, the Matrix Method combines the attractive aspects of each approach—reducing sample-to-sample contamination by using single tubes, while also increasing efficiency in terms of personnel time and cost using modified tubes that fit existing 96-well automated infrastructure and pipelines (16). Another indirect benefit of this format is that it may alleviate the need to randomize samples across each plate, although future studies should test this formally. We reiterate that since both the Matrix Method and Plate-based Method use the MagMAX Microbiome Ultra Kit extraction reagents (see Materials and Methods), this study is not an explicit comparison of DNA extraction kits. However, our previous work demonstrated the advantages of the MagMAX Microbiome Ultra Kit over other popular kits used in the field (17, 37). Future work should consider modifying the current Matrix Method to use reagents from other extraction kits that have demonstrated potential, perhaps for certain sample types not considered in our previous benchmarking of the MagMAX Microbiome Ultra Kit (17, 37).

We also attempted to validate the use of 95% isopropanol as a sample storage buffer prior to extraction for microbiome and metabolome studies. Although results based on DNA yield, sequenced read counts, and microbial community alpha- and beta-diversity confirm that storage in isopropanol generates comparable results compared to storage in 95% ethanol (Fig. 1 and 2; Fig. S1 to S5), our analyses of microbial taxon bias revealed that for the majority of sample types, storage in isopropanol reduces recovery of certain taxa (Fig. 4 to 7). We suspect that this is due to the different chemical properties that affect the ability of each solution to serve as a proper buffer for live cells and other microbial propagules. Future use of isopropanol vs ethanol as a sample storage buffer should consider these findings in order to optimize recovery of taxa. We also note that in part due to time constraints and a desire to compare storage buffers, we did not include fresh samples to compare against those stored in either isopropanol or ethanol. Previous work has shown that storage in 95% ethanol generates comparable microbiome results to fresh samples (27), but isopropanol has not been included. Future work should also investigate the differences between stored samples and fresh ones in order to see which solution best compares.

As previous studies have shown, we confirm that 95% ethanol is a suitable storage solution for microbiome studies of a diversity of sample types (27). We also confirmed that 95% ethanol effectively stabilizes the fecal metabolome at room temperature,

extending its utility for remote and at-home collection. The strong correlation between ethanol and isopropanol across all samples further supports the viability of isopropanol as a substitute, especially when navigating shipping constraints or regulatory limitations. Future work should explore the generalizability of these findings across additional sample types, time points, and preservation durations, as well as further refine solvent choice for optimal metabolite recovery. However, our results strongly support adoption of the Matrix Method as a standard approach for high-throughput, integrated microbiome studies.

## MATERIALS AND METHODS

### Sample collection and storage

Sample types, including built environment surfaces, human skin, the human oral cavity, and feces, were used to validate this proposed method due to their importance in human and environmental microbiome studies. The human subject work conducted here was approved by the University of California, San Diego (IRB#150275 for saliva and IRB#141853 for feces). In addition, informed consent has been obtained from the participants involved. Specifically, we included swabs of computer keyboards and floor tiles, human skin from the hand and armpit, human saliva from before and after brushing, and feces from both mice and humans. We used the "Earth Microbiome Project (EMP) in a box," protocols for sample collection (18), which were drafted for widespread use in benchmarking and similar studies. Twelve replicates were collected from up to four subjects for each sample, enabling inclusion of technical replicates ($n = 3$) for each of the four storage buffer-extraction method combinations (17, 36, 37) (Table 1).

Built environment samples were collected from two surfaces: computer keyboards and floor tiles. Keyboard samples involved swabbing the keys of four individual laptop computers. Floor tile samples were obtained by swabbing a 1 ft$^2$ area from tiles located in four distinct areas within a research facility at UC San Diego. Skin samples were collected from the armpits and hands of four human subjects. To reduce variability and minimize potential bias, all replicate swabs for a given sample type were handled and used simultaneously during the swabbing process. Approximately 50 mg of material was collected per built environment swab. Human oral samples included 4 mL of saliva, collected from three individuals by passive drooling into a 50 mL centrifuge tube using a funnel. These samples were then vortexed to ensure homogenization, and 200 µL was used for each extraction. Mouse fecal pellets were collected from four individuals using sterile technique. Human feces were collected from four individuals following the American Gut Project (AGP) sample collection method, which involves the swabbing of material left behind on bathroom tissue (38). Approximately 100 mg of material was collected per human fecal swab. To compare isopropanol with ethanol in terms of use as a storage buffer, all samples were immediately preserved in 700 µL of 95% (vol/vol) ethanol (25, 26) or 95% (vol/vol) isopropanol and kept at room temperature for one week prior to extraction. This was to mimic the AGP protocol of shipping samples by mail at room temperature to accommodate citizen science and ease of participation (38). We included Katharoseq controls for controlling 16S sequence-based artifacts in our low-biomass samples (33, 39). Negative controls consisted of 12 extraction blanks, one in each column of each method.

### Sample collection and storage: Plate-based Method

For the Plate-based Method, all samples were processed to have aliquots in sterile 1.5 mL polypropylene tubes (Cat#: 89004-294, VWR, PA, USA) containing 700 µL of either 95% ethanol or 95% isopropanol as a storage buffer. All samples representing built environment surfaces, human skin, and feces from humans were collected using sterile dual-headed cotton swabs (BD Falcon Swube Applicators, Cat#: 14-959-75, BD Falcon, NJ, USA). For those sample types, each swab head was broken off into a 1.5 mL tube for storage and subsequent DNA extraction. For mouse fecal samples, fecal pellets were

collected and added to each 1.5 mL tube using forceps, which were dipped in alcohol and flame sterilized between each collection. All samples were collected at the same time of day to account for circadian effects on microbiome composition. For human saliva samples, 200 µL of saliva was added to each 1.5 mL tube. For each sample, a unique sample name was designated and manually entered into a 96-well plate map for sample tracking.

### *Sample collection and storage: Matrix Method*

For the Matrix Method, all samples were processed to have aliquots in 1 mL matrix tubes (Cat#: 3740, ThermoFisher Scientific, MA, USA) containing 700 µL of either 95% ethanol or 95% isopropanol as a storage buffer. All samples representing built environment surfaces, human skin, and feces from humans were collected using custom-made sterile swabs (Affordable Solutions IHC, PA, USA) designed to fit into the matrix tubes (16). These swabs contain an easy break point, a polyester flocked head, and a high-density polypropylene (HDPP) handle. No glues or any other chemicals were used for flocking or manufacturing. Therefore, these swabs do not contribute any additional polymer leaching, unlike the cotton swabs used in the Plate-based Method (see Metabolite Extraction). For those sample types, each swab head was broken off into a 1 mL matrix tube for storage and subsequent extraction. For mouse fecal samples, one pellet was added to each matrix tube using forceps, which were dipped in alcohol and flame sterilized between each collection. For human saliva samples, 200 µL of saliva was added to each matrix tube. Since the Matrix Method facilitates both DNA and metabolite extraction on a single tubed sample, there is no need to split the sample or collect biological replicates for each extraction. All matrix tubes were assembled into 96-tube racks. These racks were then scanned using the VisionMate high-speed barcode reader (Cat#: 312800, ThermoFisher Scientific, MA, USA) to automatically generate a plate map for sample tracking.

## Metabolite extraction

As noted above, the swabs used in the Matrix Method do not contribute any additional polymer leaching, unlike the cotton swabs used in the Plate-based Method (data not shown). Similarly, the wooden handles on the swabs used in the Plate-based Method were found to absorb large quantities of each storage solution, which reduced the final volume available for sample storage. Because the excess polymers extracted and storage solution absorbed by the Plate-based Method make a fair comparison with the Matrix Method difficult, metabolomics was only performed for the Matrix Method. All samples that had DNA extracted using the Matrix Method were also subjected to untargeted metabolomics analysis to assess molecular composition and compare storage solutions for that extraction protocol. We chose untargeted metabolomics based on liquid chromatography–mass spectrometry (LC-MS/MS) in the positive ionization mode for this study due to its high sensitivity, potential to detect a broad range of molecules, and the large number of publicly accessible reference spectra.

After one week, the matrix tube rack was loaded into the SpexMiniG plate shaker at 1,200 rpm for 2 min, followed by centrifugation at 2,700 $\times$ $g$ for 5 min. The tubes were de-capped by the automated instrument, Capit-All (ThermoFisher Scientific, MA, USA). Using an 8-channel pipette and sterile technique, each sample was transferred to a metabolomics plate column by column. The metabolomics plate was stored at −80°C until direct loading into a quadrupole-Orbitrap mass spectrometer. The pellets remaining in the matrix tubes were further processed for DNA extraction, as explained below.

## DNA extraction

Both the Plate-based Method and Matrix Method use a 96-sample magnetic bead cleanup format with the KingFisher Flex and the MagMAX Microbiome Ultra Nucleic Acid Isolation Kit (Cat#: 5400630 & A42357, Thermo Fisher Scientific, MA, USA).

## DNA extraction: Plate-based Method

After one week of storage at room temperature, swabbed skin, surface, and human fecal samples were aseptically transferred from 1.5 mL polypropylene tubes into individual wells of a 96-deep-well bead plate containing zirconia beads (Cat#: A42357, Thermo Fisher Scientific, MA, USA) using sterile forceps. A residual storage buffer was left behind in the storage tube. The forceps were dipped in alcohol and flame-sterilized between each transfer to prevent cross-contamination. Mouse fecal samples preserved in 95% ethanol became too soft to transfer using forceps. To facilitate transfer, the tubes were centrifuged at 2,700 × g for 5 min, the storage buffer supernatant was discarded, and the resulting pellet was collected by swabbing. The swab head was then broken off into a single well of the previously mentioned 96-deep-well bead plate. Saliva samples were similarly centrifuged at 2,700 × g for 5 min to separate the storage buffer from the sample. Seven hundred microliters of supernatant were removed, and the remaining saliva was then transferred into the bead plate. DNA extractions were performed on this plate following the manufacturer's instructions, with lysis performed using a TissueLyser II (Cat#: 9003240, Qiagen, CA, USA), and bead clean-ups performed using the automated KingFisher Flex Purification System (Cat#: 5400630, ThermoFisher Scientific, MA, USA). Extracted DNA was stored at −80°C before quantification of DNA yield and subsequent sequencing.

## DNA extraction: Matrix Method

The matrix tubes were loaded to a SpeedVac at 45°C for 60 min at 5.1 Torr to remove any residual storage buffer. We note that with larger volumes or in rare cases, concentrating isopropanol can form peroxides, which can be hazardous. Therefore, we recommend peroxide testing of existing stocks or regularly replacing stocks for safety. Next, 30 µL of zirconia-silica beads of three distinct sizes (0.1, 0.5, and 1 mm) were added to each matrix tube using a LabTie bead dispenser (Molgen, Veenendaal, Netherlands). Then, 600 µL of lysis buffer was added to each tube using the Biotek Multiflo bulk reagent dispenser. The matrix tubes were capped using the Capit-All. Bead-beating was performed on the SpexMiniG for 2 min at 1,200 rpm. The remaining steps of the DNA extraction followed the manufacturer's instructions with bead clean-ups performed using the automated KingFisher Flex Purification System (Cat#: 5400630, ThermoFisher Scientific, MA, USA). Extracted nucleic acids were stored at −80°C before quantification of DNA yield and subsequent sequencing.

## LC-MS/MS data acquisition and processing

Untargeted metabolomics analysis was performed using an ultra-high performance liquid chromatography system (Vanquish, Thermo Scientific, MA, USA) coupled to a quadrupole-Orbitrap mass spectrometer (Q Exactive, Thermo Scientific, MA, USA). Briefly, samples were chromatographed on a Phenomenex Kinetex C18 column (1.7 um particle size, 2.1 mm × 50 mm) with flow rate of 0.5 mL/min. Chromatography solvents were as follows: Solvent A: water + 0.1% (vol/vol) formic acid; Solvent B: ACN + 0.1% (vol/vol) formic acid. Samples (5 µL injection volume) were run on a 10-min gradient with the following elution profiles: 5% solvent B for 1 min, a linear gradient from 5% to 100% solvent B in 7.5 min, 100% to 5% solvent B in 0.5 min, and 5% solvent B for 2 min. Data were collected in a data-dependent fashion in positive mode. Full mass spectrometry resolution was set to 35,000 with automatic gain control target of $5 \times 10^5$. The scan range was 100–1,500 m/z for precursor ions. For MS/MS analysis, the resolution was set to 35,000 with automatic gain control target of $5 \times 10^5$. Stepped normalized collision energy levels were 20, 30, and 40. The minimum automatic gain control target was $5 \times 10^3$. The apex trigger was set to 2–15 s with dynamic exclusion of 10 s.

Thermo proprietary MS files (.raw) were converted to a GNPS-compatible format (.mzML) using the Proteowizard program MSConvert and processed using MZmine3 for feature finding. Data were batch-processed and filtered by assigning the advanced MS[1]

and MS² detector threshold with a noise factor of 3 in the factor of the lowest signal algorithm in the mass detection module. The following parameters were applied to extract mass spectrometric features:

(i) ADAP chromatogram builder (minimum intensity of the highest data point in the chromatogram, 2E5; m/z tolerance, 10 ppm)
(ii) Chromatogram deconvolution (chromatographic threshold of 85%; minimum absolute height, 2E5; minimum ratio of peak top/edge, 1.4; m/z range for MS2 scan pairing, 10 ppm; retention time tolerance for MS² scan pairing, 0.05 min)
(iii) 13C isotope filter (m/z tolerance, 5 ppm; retention time tolerance, absolute, 0.04 min; maximum charge, 2; representative isotope, most intense)
(iv) Join aligner (m/z tolerance, 10 ppm; retention time tolerance, 0.15  min)
(v) Feature list rows filter (minimum features in an isotope pattern, 2; never remove feature with MS²)
(vi) Remove duplicate filter (retention time tolerance, absolute, 0.03 min; m/z tolerance, three ppm)
(vii) Peak finder (intensity tolerance, 0.1; retention time tolerance, absolute, 0.1 min; m/z tolerance, 10 ppm)

The two output files from MZmine3 were a feature table with ion intensities (.csv file format) representing the MS¹ feature information and a corresponding list of MS² spectra linked to the MS¹ features (.mgf file format). We performed *in silico* compound annotation using SIRIUS version 6.2.2 (40), including the following adducts: $[M + H]^+$, $[M + Na]^+$, $[M + K]^+$, $[M + NH_4]^+$, and $[M-H_2O + H]^+$. Molecular formula determination was performed using the built-in fragmentation tree-based approach. For chemical classification and structure prediction, we employed the CSI:FingerID module, which matches experimental MS/MS data against molecular fingerprints predicted from structure databases. All parameters were used with default settings unless otherwise specified. Mass spectrometry data generated in this study is available publicly in MassIVE under the accession number MSV000090083. Following removal of singleton features, the final feature table for diversity analyses included 154 samples and 1,165 metabolites.

## 16S and shotgun metagenomics sequence data generation and processing

We prepared DNA for 16S (V4) and shallow shotgun metagenomics sequencing as described previously (41–45). For 16S sequence data, raw sequence files were demultiplexed, reads trimmed to 150 bp, and 100% operational taxonomic units (i.e., subOTUs) were generated using the Deblur pipeline in Qiita (46, 47). We then performed a series of processing steps using QIIME 2 (version: amplicon-2024.10) (48): reads were filtered to align with the Greengenes 2 (version: 2024.9) database (49), with additional taxonomic assignments to the SILVA 138.2 database (50) in order to classify plastid sequences not yet included in Greengenes 2. Mitochondrial, chloroplast, eukaryotic, and unassigned sequences were excluded, as well as those bacterial sequences with no classification below the domain level. Control samples were excluded, and then high- and low-biomass samples were filtered into separate feature tables. Next, we removed singleton features from each data set and created normalized tables for diversity analyses. For alpha-diversity and the majority of beta-diversity analyses (i.e., those for Jaccard, unweighted UniFrac, and weighted UniFrac distances) (51), the low-biomass data set was rarefied to 277 reads per sample, and the high-biomass data set to 20,636 reads per sample. Rarefaction depths were selected to exclude samples below our Katharoseq minimum threshold (33) and to maintain at least 75% of samples within each sample type (19). For beta-diversity analyses using Robust Aitchison Principal Components Analysis (RPCA) (52) or its phylogenetic counterpart, phylo-RPCA (53), samples with fewer than 277 reads were excluded from the low-biomass data set, whereas those with fewer than 20,636 reads were excluded from the high-biomass data set. The final feature table used for analyses of low-biomass samples included 112 samples and 1,331

taxa (1,742 for RPCA/phylo-RPCA), and the table for high-biomass samples included 108 samples and 1,720 taxa (1,730 for RPCA/phylo-RPCA).

For shotgun metagenomics data, raw sequence files were demultiplexed using BaseSpace (Illumina, CA, USA) and subsequently quality- and host-filtered (54). Per-sample, quality-filtered reads were uploaded to Qiita for the generation of operational genomic units (OGUs) using the Web of Life Toolkit App (Woltka) (55) (Qiita-specific version: 0.1.7 paired-end). Qiita's Woltka process aligns reads to either the Web of Life database (56) or a curated NCBI database of microbial genome (RS225), using bowtie2 (57). Settings included maximum and minimum mismatch penalties (mp = [1, 1]), a penalty for ambiguities (np = 1; default), read and reference gap open and extend penalties (rdg = [0, 1], rfg = [0, 1]), a minimum alignment score for an alignment to be considered valid (score-min = [L, 0, −0.05]), a defined number of distinct, valid alignments (k = 16), and the suppression of SAM records for unaligned reads, as well as SAM headers (no-unal, no-hd). Resulting alignments are converted to a feature table. The version of the Web of Life database used contained 15,953 archaeal and bacterial genomes. The RS225 database used is a collection of reference microbial genomes sampled from the NCBI RefSeq genome database (i.e., release 225) and contains 40,987 genomes from NCBI RefSeq and 11,771 genomes from external sources. These genomes span four domains and represent 32,216 species, including 610 fungal genomes. The database provides genomes, lineage information, and a pre-built host-depleted index, all of which are publicly available at: https://ftp.microbio.me/pub/RS225/. We used the Web of Life database to characterize Archaea and Bacteria, harnessing its phylogeny for our microbial diversity analyses and the RS225 database for characterizing fungi for our analyses of recovery of fungal taxa.

For Archaea and Bacteria, as for 16S data above, following the removal of control samples, singleton features were removed, and the data normalized for diversity analyses. For alpha-diversity and the majority of beta-diversity analyses, the low-biomass data set was rarefied to 55,892 reads per sample, and the high-biomass data set to 1,515,275 reads per sample. Rarefaction depths were selected to maintain at least 75% of samples within each sample type. For beta-diversity analyses using RPCA/phylo-RPCA, samples with fewer than 55,892 reads were excluded from the low-biomass data set, and those with fewer than 1,515,275 reads were excluded from the high-biomass data set. The final feature table used for analyses of low-biomass samples included 98 samples and 5,635 taxa (6,701 for RPCA/phylo-RPCA) and the table for high-biomass samples included 109 samples and 6,413 taxa (6,815 for RPCA/phylo-RPCA). For fungi, following the removal of control samples, non-fungal features were removed, and then, singleton features were excluded. The final feature table used for assessing recovery of fungal taxa from low-biomass samples included 119 samples and 26,361 features, and the table for high-biomass samples included 144 samples and 17,928 features.

## LC-MS/MS metabolomics data analysis

For analysis of our metabolomics data, we focused on the effect of storage solution (i.e., isopropanol vs ethanol) for only samples extracted using the Matrix Method (see Metabolite Extraction). To compare alpha-diversity between storage solutions, we quantified richness for each sample using QIIME 2 and modeled the relationship between storage solutions using R (58). We assessed results from both Spearman and Kendall correlations, as well as those from a two-sided, paired $t$-test. To compare beta-diversity, we used QIIME 2 to run Mantel correlations (59) between storage solutions, separately for each sample type, and assessed results for each of the Jaccard, Bray–Curtis (60), RPCA, Canberra–Adkins (61), and cosine dissimilarities/distances. To further assess the influence of storage solution on beta-diversity of metabolites for each sample type, we visualized within-sample distances among technical replicates for each sample type and quantified differences among storage solutions using Wilcoxon rank sum tests in R. We also used QIIME 2 to run dispersion analysis (PERMDISP) and permutational analysis of variance (PERMANOVA) (62) to compare the effects of

host subject and storage solution on beta-diversity for each sample type and for each distance metric described above. To better understand how metabolites varied between storage solutions for each sample type (i.e., metabolite bias), we visualized the total number of metabolites recovered, as well as the number of metabolites exclusive to each method, using UpSet plots (63). All non-singleton features were considered for these analyses. To determine which groups of metabolites varied most, we binned individual features by subclass and visualized patterns using a heatmap. Specifically, intensities were summed across all non-singleton features within each subclass and then log-transformed for visualization.

## 16S and shotgun metagenomics data analysis

For analysis of our microbiome data, we were able to compare storage solutions within and between extraction methods. We quantified DNA yield using the Quant-iT PicoGreen dsDNA assay (Cat#: P7589; Thermo, MA, USA). We first compared differences in DNA yield among protocols using boxplots and Kruskal–Wallis tests, ignoring the paired nature of our sampling, as input amounts were kept consistent during sampling but were not quantified or normalized based on mass or volume. Therefore, for any sample type, we did not expect replicates of the same sample to have more consistent DNA yields compared to those from different samples. Still, we modeled the relationship between replicates as above for metabolite alpha-diversity, but using one-sided $t$-tests when comparing extraction methods, as we expected higher DNA yields from the larger swabs used in the plate-based method. We focused our diversity analyses on Archaea and Bacteria to harness the Web of Life phylogeny, as there is no currently comprehensive phylogeny that includes Fungi. For alpha-diversity, we quantified Faith's phylogenetic diversity (PD) (64) for each sample using QIIME 2 and modeled the relationships between replicates as above to make comparisons among storage solutions and extraction methods. We compared beta-diversity using Mantel tests as above for metabolites, but used Jaccard, RPCA, unweighted UniFrac, weighted UniFrac, and phylo-RPCA distances. To further assess the influence of storage solution and extraction method on beta-diversity of archaeal and bacterial taxa for each sample type, we visualized within-sample distances among technical replicates for each sample type and quantified differences among storage solutions using Kruskal–Wallis tests in R. We also used QIIME 2 to run PERMDISP and PERMANOVA to compare the effects of host subject, storage solution, extraction method, and the interaction between storage solution and extraction method on beta-diversity, for each sample type and for each distance metric described above. To assess taxon bias for Archaea, Bacteria (species-level), and Fungi (genus-level), we focused on our shotgun metagenomics data, considering all non-singleton features for these analyses. To better understand which archaeal and bacterial taxa varied among protocols for each sample type, we visualized the total number of taxa recovered, as well as the number of taxa exclusive to each method, using UpSet plots and a phylogeny-based approach, Empress (65). To determine which fungal genera varied most, we visualized read counts using a heatmap, as above for metabolites.

## ACKNOWLEDGMENTS

This work was supported by the Alzheimer's Gut Microbiome Project, grant number U19AG063744. This publication includes data generated at the UC San Diego IGM Genomics Center, University of California, San Diego, La Jolla, CA, utilizing an Illumina NovaSeq 6000 that was purchased with funding from a National Institutes of Health SIG grant (#S10 OD026929).

## AUTHOR AFFILIATIONS

[1]Department of Pediatrics, University of California San Diego, La Jolla, California, USA
[2]Division of Biological Sciences, University of California San Diego, La Jolla, California, USA

[3]Department of Biology, California State University, Fresno, California, USA

[4]Collaborative Mass Spectrometry Innovation Center, Skaggs School of Pharmacy and Pharmaceutical Sciences, University of California San Diego, San Diego, California, USA

[5]Bioinformatics and Systems Biology Program, University of California San Diego, La Jolla, California, USA

[6]Department of Bioengineering, University of California, San Diego, La Jolla, California, USA

[7]Center for Microbiome Innovation, University of California, San Diego, La Jolla, California, USA

[8]Department of Computer Science and Engineering, University of California, San Diego, La Jolla, California, USA

## AUTHOR ORCIDs

Caitriona Brennan  http://orcid.org/0000-0003-3943-6701

Justin P. Shaffer  http://orcid.org/0000-0002-9371-6336

Pedro Belda-Ferre  http://orcid.org/0000-0001-6532-1161

Ipsita Mohanty  http://orcid.org/0000-0001-5311-6443

Yuhan Weng  http://orcid.org/0009-0005-3453-1402

Gail Ackermann  http://orcid.org/0000-0002-3901-4931

Celeste Allaband  http://orcid.org/0000-0003-1832-4858

MacKenzie Bryant  http://orcid.org/0000-0003-0749-2995

Sawyer Farmer  http://orcid.org/0009-0006-4160-9535

Daniel McDonald  http://orcid.org/0000-0003-0876-9060

Michael J. Meehan  http://orcid.org/0000-0002-6836-0294

Gibraan Rahman  http://orcid.org/0000-0002-8843-0229

Rodolfo A. Salido  http://orcid.org/0000-0003-0631-2817

Se Jin Song  http://orcid.org/0000-0003-0750-5709

Caitlin Tribelhorn  http://orcid.org/0009-0002-8788-471X

Helena M. Tubb  http://orcid.org/0000-0002-7637-5090

Pieter C. Dorrestein  http://orcid.org/0000-0002-3003-1030

Rob Knight  http://orcid.org/0000-0002-0975-9019

## FUNDING

| Funder | Grant(s) | Author(s) |
| --- | --- | --- |
| Professors of the Future | SD IRACDA - Professors of the Future - 5K12GM068524-17 | Rob Knight |
| Common Fund | U19AG063744, 1DP1AT010885 | Rob Knight |

## AUTHOR CONTRIBUTIONS

Caitriona Brennan, Conceptualization, Data curation, Formal analysis, Investigation, Methodology, Project administration, Validation, Writing – original draft, Writing – review and editing | Justin P. Shaffer, Conceptualization, Data curation, Formal analysis, Investigation, Methodology, Project administration, Validation, Writing – original draft, Writing – review and editing.

## DATA AVAILABILITY

All data have been made publicly available at the EMBL EBI European nucleotide archive (accession number ERP141755) and through Qiita (Qiita study ID: 14332). Mass spectrometry data generated in this study is available publicly in MassIVE under the accession number MSV000090083. All input data files and analysis code are available on GitHub (https://github.com/justinshaffer/matrix_etoh_ipa).

## ADDITIONAL FILES

The following material is available online.

### Supplemental Material

**Supplemental figures and tables (Spectrum01912-25-s0001.docx).** Figures S1 to S7 and Tables S1 to S3.

### Open Peer Review

**PEER REVIEW HISTORY (review-history.pdf).** An accounting of the reviewer comments and feedback.

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
