## [Reviewer comments · Microbiology Spectrum]

Microbiology Spectrum

Streamlined Extraction of Nucleic Acids and Metabolites from Low- and High-Biomass Samples Using Isopropanol and Matrix Tubes

Caitriona Brennan, Justin Shaffer, Pedro Belda-Ferre, Ipsita Mohanty, Yuhan Weng, Kalen Cantrell, Gail Ackermann, Celeste Allaband, MacKenzie Bryant, Sawyer Farmer, Antonio González, Daniel McDonald, Cameron Martino, Michael Meehan, Gibraan Rahman, Rodolfo Benitez, Tara Schwartz, Se Jin Song, Caitlin Tribelhorn, Helena Tubb, Pieter Dorrestein, and Rob Knight

Corresponding Author(s): Rob Knight, University of California San Diego

Review Timeline:

Submission Date:	September 10, 2025
Editorial Decision:	September 12, 2025
Revision Received:	September 12, 2025
Accepted:	September 15, 2025

Editor: Jan Claesen

Reviewer(s): The reviewers have opted to remain anonymous.

Transaction Report:

DOI: <https://doi.org/10.1128/spectrum.01912-25>

Re: Spectrum01912-25 (Streamlined Extraction of Nucleic Acids and Metabolites from Low- and High-Biomass Samples Using Isopropanol and Matrix Tubes)

Dear Prof. Rob Knight:

Thank you for the privilege of reviewing your work. Below you will find my comments and instructions from the Spectrum editorial office.

I am pleased to inform you that your manuscript has been editorially accepted for publication. However, there are a few additional questions in the submission form (specific to Microbiology Spectrum) that need to be answered before the final decision. Once these are completed, please return your submission so that I can move your paper forward to acceptance.

Revision Guidelines

Sincerely,
Jan Claesen
Editor
Microbiology Spectrum

Re: mSystems01206-22 (All in one, rapid, low cost and high-throughput method for microbiome sample accession, DNA and metabolite extraction)

Dear Dr. Rob Knight:

After considering detailed feedback from two reviewers and after consultation with two senior editors, I regret to inform you that we will not be able to publish your manuscript entitled "All in one, rapid, low cost and high-throughput method for microbiome sample accession, DNA and metabolite extraction" in mSystems. The reviewers appreciated your desire to share new methods that can increase the efficiency of microbiome sample processing while also reducing the cost. But there are many major issues identified in the review that make the work unsuitable for mSystems (see comments below from reviewers).

While this manuscript may not be a good fit for mSystems, please note that you do have the option to transfer this manuscript to Microbiology Spectrum. Microbiology Spectrum is an open-access journal from the ASM that seeks to publish technically sound, primary research across the entire range of microbial sciences and allied fields. More information on Microbiology Spectrum can be found here: <https://journals.asm.org/journal/spectrum>. As your article was reviewed at mSystems, you can transfer the paper along with the reviews and reviewer identities to Microbiology Spectrum. The Spectrum Editors will carefully assess your response to technical concerns raised in peer review in their decision-making process. **Please note that Spectrum only publishes primary research. Spectrum does NOT currently publish or consider case studies, reviews, meta-reviews, commentaries, opinions/hypotheses, perspectives, or mini-reviews.**

You may transfer your paper to Microbiology Spectrum by clicking on the link below. Please note that the transfer link below will be visible only in the decision letter sent to the corresponding author.

Please reach out to the Spectrum journals team (spectrum@asmusa.org) if you need any further information prior to transferring your paper to Microbiology Spectrum.

If you would like to transfer the manuscript and the mSystems reviews to Spectrum, please use this link:

Link Not Available

Thank you for considering mSystems.

Sincerely,

Benjamin Wolfe
Editor, mSystems
mSystems@asmusa.org

Reviewer comments:

Reviewer #1 (Comments for the Author):

This manuscript presents a modified version of a previous "all-in-one" microbiome sample prep method that can help reduce some of the steps required. I appreciate that the authors want to make microbiome sample prep cheaper, easier, and more high-throughput. I also appreciate that they want to compare their previous method to this new commercially available sample prep tube from ThermoFisher. But I did not find the work to provide a major methodological advance that would be widely used by many labs. Moreover, the manuscript was very poorly formatted and presented for review, making it challenging to read and understand. It was also very underdeveloped in terms of providing the reader with clear and nuanced writing that fully unpacked the details and rationale of this approach.

Response: We are thankful for these helpful comments. Based on these reviews, we have made major changes to our presentation of this material, such as inclusion of data from additional sample types. As part of this effort, we published an observation paper in *mSystems* focusing on a distinct, smaller dataset including only human feces: <https://doi.org/10.1128/msystems.00985-24>, and which demonstrates that the Matrix method can reduce time required from technicians, sample-to-sample contamination, and overall processing time for samples (i.e., some of the major points of our initial manuscript). The updated manuscript we are submitting here expands upon our initial submission and this observation by focusing on comparison of sample storage buffers (i.e., isopropanol vs. ethanol), and differences in taxon bias between DNA extraction methods. We included data from mouse feces, human feces, the human oral cavity, human skin, and the built environment. To highlight major advances noted to be missing in our initial draft, we showcase use cases for isopropanol vs. ethanol as a storage solution, as well as major improvements in taxon recovery when using the Matrix method vs. a more traditional 96-well plate-based one. Finally, we have re-written the entire paper to improve formatting and overall presentation in part to better deliver the details and rationale of each method. The changes occur throughout, and we note where specific changes occur when relevant below.

It is standard practice to include both page and line numbers in manuscripts that are submitted for review. Both were lacking, making it hard for this reviewer to provide advice on specific lines/paragraphs sections.

Response: We have added page and line numbers as requested. The changes occur throughout.

One of the biggest issues that I see with the current manuscript is that it is presented as something that should be widely adopted, but is written as a modification to an in-house protocol for one or two lab groups at UC San Diego. No other widely used DNA extraction protocols were compared to this new protocol, so it is difficult to know how it would compare to other widely used methods. This lack of a broader comparison of methods was surprising giving that the authors just did something similar in a different study: <https://www.future-science.com/doi/full/10.2144/btn-2022-0032> Moreover, I don't expect most labs to be able to use this approach given the expense and advanced equipment needed. The cost per sample prep is cheaper, but it wasn't clear to me that they took into account the expensive equipment (like a

\$15,000 barcode reader) that would be needed for handling the tubes. This was never pointed out in the manuscript.

Response: We are thankful for these helpful comments. In our updated manuscript, we have clarified that the in-house protocol is indeed a commercially-available and widely used kit from Thermo (MagMAX Microbiome), and that a major goal of the paper is to compare the Matrix method with that plate-based approach. We have also rewritten the paper to avoid suggestions that the Matrix method should be widely adopted without also discussing the cost of implementing the pipeline including equipment costs. The changes occur throughout, but specifically on page 4.

Only a very limited number of samples were used for testing out this protocol: 3 saliva, 4 mouse feces, and 4 human feces. There can be a huge range of taxonomic diversity in all of these sample types that is not captured by this limited number of samples. Without this range of taxonomic diversity, we cannot fully understand biases in this approach, taxa that may be under/over-represented, etc. These data are preliminary and limited in scope and do not make a comprehensive and complete methods optimization paper.

Response: We appreciate this comment. Our typical approach for benchmarking these types of methods is often restricted by a ceiling of 96 samples, which is limiting but allows us to avoid assessing a plate effect in our experiments. Furthermore, our approach includes technical replication of each biological sample that allows us to focus on within- vs. between-sample variation in our analyses, which reduces the need for large sample sets for each type. Therefore, we purposefully limited the total number of samples in this way, but also balanced the number of samples per type with the number of types, to maximize scope in terms of the latter. Although somewhat restrictive, this approach has been used to benchmark the sample storage and nucleic acid extraction methods used by large-scale projects such as the Microsetta Initiative (i.e., American Gut Project) and Earth Microbiome Project, and which are also widely adopted in the field. As noted, we have included a discussion of the potential limitations of our study in terms of sample sizes per type, as well as other aspects. The changes occur throughout the discussion, on page 20 beginning on line 492.

Are there any other single tube methods out there that other researchers have presented? This was not clear to me. It would have been very helpful to provide a discussion of how those methods compare to this current method.

Response: We appreciate this comment. We have made sure to include a discussion of previously reported studies comparing single tubes to plates and similar. The changes occur on page 20, beginning on line 502.

The Results describing differences in some taxa that were detected across the two sample types was severely underdeveloped. What were the taxa that were uniquely recovered in the Matrix Method approach? Why were these taxa potentially only present in that approach? It is really hard to figure out what taxa are being shown in Figure S6.

Response: We are thankful for these comments. We have further developed the discussion of these results including the specific taxa potentially only present in one approach. The changes occur on pages 11 lines 287-305, pages 12-13 lines 321-377, and pages 17-19 lines 440-486.

The Discussion was also severely underdeveloped. There was not a discussion of potential limitations of this approach/dataset or a discussion of how it could be applied across lab groups (limited # of samples, lack of fungal data, etc.). There was also limited contextualization in terms of other methods that are currently used. There could be a helpful discussion of why some taxa were missing in one method versus another. It also did not include a discussion of how this method might perform with other non-animal microbiome sample types.

Response: We appreciate these comments. We have updated our manuscript to include discussions of potential limitations, other similar methods, exclusiveness of certain taxa, and other sample types not included here. The changes occur on page 20, beginning on line 492.

Approximately how much of each sample type (in mg or uL/mL) was extracted in each tube? That is not clear. It just says that swabs were used to collect the samples.

Response: We are thankful for this comment. We have updated our methods to include approximate amounts of each sample type when swabs were used, as suggested. The changes occur on page 23, beginning on line 570.

I was surprised to see a lack of fungal metagenomic data despite an increasing number of studies showing that fungi are important parts of gut and other human microbiomes. In fact, the authors note in the introduction that "the effectiveness of bead-beating in releasing DNA from fungal propagules is considered a better alternative due to lower cost and shorter processing time" but they do not present data on how well their new method captures fungal diversity.

Response: We are thankful for this comment. We have included an analysis of fungal taxon bias in our updated manuscript. The changes occur throughout the manuscript.

In journals where the Results come before the methods sections, it is important to do some "methodological previewing" in the Results so the reader can understand where the results are coming from. This was lacking in your Results narrative and made it very difficult to read and understand the results. For example, in the first results section, I didn't understand what you meant by "sample brushing" until I dug far down into the methods. See various recent mSystems manuscripts for examples of how to do this.

Response: We appreciate this comment. We have updated our results section to appropriately reflect to order of sections in the journal article. The changes occur throughout the results.

I found the well-to-well contamination experiments limited in design and poorly presented. First, it was not clear to me in the methods if the authors took any simple precautions to avoid well-to-well contamination. There are very simple, cheap, and quick tools available (removable foil, strip caps) that could be used to prevent contamination in plates. Only a single 96 array was used for each method type meaning there is no plate replication. This limits our ability to understand how

common the limited amount of contamination that was noted in the plate method actually occurs across multiple plates. Perhaps the plate that was analyzed was just poorly loaded. I also found Fig 3C not very effective at displaying the contamination. Perhaps just label wells where contamination was present (where DNA was over some background threshold)? It is really hard to see differences in the colors of the DNA concentration gradient.

Response: We are thankful for these comments. We have moved the analysis and discussion of well-to-well contamination to a separate manuscript that was recently published: <https://doi.org/10.1128/msystems.00985-24>. Briefly, that paper demonstrates that the Matrix Method reduces the impact of well-to-well contamination during sample preparation for DNA extraction, as compared to the same extraction protocol using a 96-well plate-based method. We note earlier how our general approach for benchmarking similar protocols purposely uses a single 96 array to avoid the need to assess an effect of plate. There are a handful of previous studies that have demonstrated the magnitude of well-to-well contamination for similar sample sets: <https://doi.org/10.2144/btn-2022-0032>; <https://doi.org/10.1128/msystems.00218-17>. This is despite using some of the quick tools suggested (removable foil, strip caps), as even the smallest volumes of sample can be spread when removing such seals from plates or tubes, as demonstrated in these three papers.

If I understand figure S6 correctly, I am surprised to see that many species were only detected in one method and not both. This is potentially problematic for making comparisons between studies that use the two methods. This figure should be in the main text given the implications of this finding.

Response: We are grateful for this comment. We have included the figures concerning taxon bias in the main text, as suggested. The changes occur throughout.

Figure S6. "Values indicate counts." Counts of what? Is this just species richness?

Response: We are thankful for this comment. This indeed species richness, and we have clarified this in the main text and figure legends for these figures. The changes occur throughout.

Reviewer #2 (Comments for the Author):

Brennan et al. presents a streamlined method of extracting microbiome samples for DNA and metabolite analysis that produces similar results as a previously benchmarked protocol while also reducing human error, processing time, and costs. The authors also compare isopropanol as a sample storage solution against ethanol within their new process. I'd like to share appreciation for this methods-sharing and development manuscript as being transparent in this area is beneficial to everyone in the field.

Response: We appreciate this comment.

Major comments:

There are several mentions of the prior Plate-based method being previously benchmarked. But, I did not see those references in the manuscript. I am curious if an extrapolation can be made to how the Matrix Method compares to Tube-based methods. Also, is it possible to compare how extracted DNA concentrations would compare to Tube-based methods as well?

Response: We are thankful for these comments. We have clarified the previous studies benchmarking the Plate-based method. We have also included a discussion of how the Matrix method compares to other tube-based methods. The changes occur on page 4 lines 84-89 and page 20 beginning on line 505.

Why were custom swabs only used for the Matrix Method and not the Plate-based method? Relatedly, Table 2 did not contain time required for adding swabs to the Matrix tubes. Is this because the expectation is that this time burden would be placed onto the participant providing the stool sample? If so, this should be made clearer.

Response: We appreciate these comments. Custom swabs were only used for the Matrix method to more accurately represent the Plate-based method, which has traditionally been benchmarked using the wooden swabs used for that method. So, by using the wooden swabs with the Plate-based method we were able to more accurately compare the Matrix method to this other approach. Also, we have moved the analysis and discussion of processing time to a separate manuscript that was recently published: <https://doi.org/10.1128/msystems.00985-24>. Briefly, that paper demonstrates that the Matrix Method reduces the time of sample processing during sample preparation for DNA extraction, as compared to the same extraction protocol using a 96-well plate-based method.

I think the cost breakdown of the Matrix Method could be fleshed out more. The cost comparison of consumables for both methods is useful (Table 3). The usefulness would be enhanced if additional tables were provided on initial, upfront costs for adopting the Matrix Method. This would include the price of instruments such as the VisionMate high speed barcode reader, the automatic capping machine, and the (custom?) LabTie bead dispenser. Without the consideration of these instruments, some time savings/error reduction may not be applicable (e.g. scanning Matrix tubes by hand rather than by instrument). This may reduce the savings of the

Matrix Method but a case could be made on how quickly the method pays itself off. I am also curious if there is a large difference in yearly instrument maintenance between the two methods.

Response: We are grateful for these comments. We have moved the comparison and discussion of cost to a separate manuscript that was recently published:

<https://doi.org/10.1128/msystems.00985-24>. Briefly, that paper demonstrates that the Matrix Method has lower consumable costs and reduced sample loss, which help offset higher capital costs, as compared to the same extraction protocol using a 96-well plate-based method.

More minor points:

Line numbers would have been useful for the review of this manuscript.

Response: Thank you for this helpful comment. We have added page and line numbers throughout.

Because this is a methods paper, I think more details of fecal collection method would be useful. I was trying to find out what the 'Earth Microbiome Project (EMP) in a box' protocol meant. I followed the citation from the manuscript to Shaffer et al. but couldn't find the phrase. I followed that Shaffer et al. to Thompson et al. where I found more detailed methods but still no mention of 'EMP in a box'. I think it would help with method clarity if that part was elaborated.

Response: Thanks for this comment. The 'Earth Microbiome Project (EMP) in box' is indeed mentioned in the first paragraph of Shaffer et al: <https://doi.org/10.2144/btn-2022-0032>. Regardless, we made sure this reference is clear, and also elaborated on the details of sample collection for clarity, in our updated manuscript. The changes occur on page 23 beginning on line 564.

I have some comments about isopropanol (IPA) as a storage solution. Firstly, the regulatory restrictions around ethanol as a motivation for investigating IPA should be further explained. Like ethanol, IPA is classified as a hazardous material for shipping. For transport, it is classified under the same hazard class and packing group as ethanol for Department of Transportation (US), International Maritime Dangerous Goods, and International Air Transport Association. As written, I'm unconvinced that the switch to IPA would make shipping samples any easier, from a regulation standpoint.

Response: We are thankful for these comments. We have clarified our justification for assessing isopropanol as an alternative storage solution. The changes occur on pages 5-6, in the final paragraph of the introduction.

The manuscript is also missing discussion about the hazard considerations of IPA over ethanol. IPA is classified as a Group B peroxide forming chemical. As such, peroxide formation tests are recommended before evaporation/condensation/distillation. The Matrix Method includes steps that involve concentrating, evaporation, or low temperatures (Fig 1A) where peroxide formation should be considered (storage at low temperatures can cause peroxides to crystalize out of

solution). Additionally, both IPA and ethanol are classified as flammable liquids and eye irritants, but IPA is also classified as having specific target organ toxicity (single exposure and central nervous system). The IPA SDS advises to avoid generation of vapors/aerosols, work under hood, and not to inhale substance. Have the researchers consulted with OESO about potential safety concerns that may need to be relayed to users?

Response: We are thankful for these comments. We have included a discussion of the potential hazards of evaporating residual volumes (<100 µL) of 95% isopropanol in terms of peroxide formation, which also includes details provided by our Environmental Health and Safety office. The changes occur on page 27, in the 'DNA Extraction: Matrix Method' paragraph.

IPA availability as a motivation could be strengthened by supporting evidence. I'm assuming the authors are referring to supply chain inconsistencies during the COVID-19 pandemic. Indeed, in the case of the pandemic, flexibility of the storage solution would be helpful. But the authors' specific argument of availability without supporting evidence is not convincing, especially given the increased safety hazards as mentioned above. The IPA supply chain was also affected by the COVID-19 pandemic as it is a major raw material in disinfectant products and hand sanitizers. I think what would help strengthen this argument is more supporting evidence for why flexibility of storage solution is preferred, perhaps of ethanol vs IPA prices over time or other supply chain considerations.

Response: Thank you. We have clarified the importance of availability on pages 5-6, in the final paragraph of the introduction.

I'm confused about mock community dilutions. They are mentioned under 'DNA Extraction' in the Methods section. However, they are missing from Table 1 and Fig 3 which make me think they do not get extracted (or maybe on a separate plate?). The ZYMOBIOMICS Microbial Community DNA Standard is mentioned in the 'Well-To-Well Contamination Experiment' but that's purified DNA intended as qPCR standard.

Response: Thanks for this comment. We have moved the comparison and discussion of well-to-well contamination, which includes these dilutions, to a separate manuscript that was recently published: <https://doi.org/10.1128/msystems.00985-24>.

If the data is available, it would be interesting to see a comparison to the previous bead-beating protocol. A paragraph in the introduction was devoted to bead-beating. It's unclear from the manuscript if bead-beating with the three different zirconia-silica bead sizes is another new method that hasn't been benchmarked yet.

Response: We appreciate this comment. Although the three sizes were used in a previous publication, that paper focused on well-to-well contamination and not taxon bias, as it focused on just a single sample type (human feces): <https://doi.org/10.1128/msystems.00985-24>. Therefore, we have included a more in-depth discussion of the potential benefits of the three bead sizes. The changes occur throughout, but specifically on pages 17-18 beginning on line 429.

In the instances where storage solution is a major explanatory factor (e.g. mouse feces), it would

have been helpful to have a gold standard control (analysis of fresh, unpreserved samples). I presume such a control is not possible at this point (but it would be great to incorporate if such data do exist!) In the case where it is not possible to go back and redo the experiment, is there any way of guessing which Method might have been closer to the "true" answer? Such speculation would be useful for researchers considering these techniques.

Response: We appreciate this comment. Due to this work being part of a PhD dissertation, it is not possible to go back and redo the experiment to include fresh samples. Furthermore, the breadth of sample types included prevents fresh samples from being assessed together, as even short amounts of storage are required between collections of the different types. In previous studies comparing storage solutions with fresh samples, ethanol was found to be most comparable, however isopropanol was not considered in that study: <https://journals.asm.org/doi/10.1128/msystems.01329-20>. We have added a discussion of this/fresh samples to our updated manuscript. The changes occur on page 22 beginning on line 536.

Why metabolomics was unsuccessful with the Plate-based Method if it was previously benchmarked with that method? Could the authors elaborate on this point?

Response: We appreciate your comment. Our experimental design involved the storage of samples for one week at room temperature. Under these conditions, the full 700 μL of storage solution was absorbed by the wooden swab in each Plate-based sample, leaving insufficient solvent for metabolite extraction and thereby precluding metabolomic analysis. However, our recent publication demonstrates that the Matrix Method using 95% ethanol effectively recovers metabolite profiles comparable to those obtained with a validated and commonly used 50% methanol extraction, supporting its robustness for metabolomic applications: <https://pubs.acs.org/doi/10.1021/acs.analchem.4c05142>. We have clarified this on page 8 lines 181-183, on page 19 lines 461-467, and on page 25 lines 625-631.

I would be interested in whether randomization to reduce batch effects is still required using the Matrix Method. Perhaps this could be a discussion point in the manuscript.

Response: We appreciate this comment. We have added a discussion of whether randomization is still needed when using the Matrix Method to our updated manuscript. The changes occur in page 21 beginning on line 518.

Table S1: Why are some factors missing? For example, 16S, Human Feces, Weighted Unifrac: Lacks statistics associated with extraction_protocol, storage_solution. Also, for the tests with lower host_subject_ID effects here, can the authors speculate on why that may be the case? Is it low biomass of input samples?

Response: Thanks for this comment. Some factors were missing from that analysis because it only includes those with an effect above a certain threshold – all other variables are dropped from the model. However, we have excluded this analysis in our updated manuscript in favor of PERMDISP and PERMANOVA, which include results for all factors. The changes occur throughout the results.

Grammar and formatting:

There is a reference in parentheses in the Results to "Pearson's $r > 0.64$ ". Based on the context, it seems that should be a "Mantel" instead?

Response: We appreciate this comment. We have clarified that the correlation coefficient is from a Mantel test. The changes occur throughout the results.

It seems a bit odd to say later that "Mantel correlations prove strong correlations." I recommend rephrasing to sound less circular.

Response: We are thankful for this comment. We have updated the text to be less circular as suggested. The changes occur throughout the results.

In the 'Sample Collection and Storage: Matrix Method' section, there's a sentence that begins with "Each human fecal swab..." where "1ml" should be changed to "1 mL" to match the formatting of the rest of the manuscript. The same formatting discrepancy occurs in the Plate-based Method table of Table 3.

Response: We are grateful for this comment. We have normalized the formatting of units as suggested. The changes occur throughout the manuscript.

In the last sentence of the 'DNA Extraction: Plate-based Method' section, the degree symbol used in '-80{degree sign}C' is incorrect.

Response: We are grateful for this comment. We have corrected this degree symbol as suggested.

In 'Step 1' of FIG 1 (A), isopropanol is incorrectly capitalized. In 'Step 6' of the same figure, isopropanol should be mentioned as well.

Response: We are thankful for these comments. We have moved this diagram to a separate manuscript that was recently published: <https://doi.org/10.1128/msystems.00985-24>.

Re: Spectrum01912-25R1 (Streamlined Extraction of Nucleic Acids and Metabolites from Low- and High-Biomass Samples Using Isopropanol and Matrix Tubes)

Dear Prof. Rob Knight:

Thanks for addressing the last few Spectrum-specific items in the submission portal. Your paper is now officially accepted for publication!

Your manuscript has been accepted, and I am forwarding it to the ASM production staff for publication. Your paper will first be checked to make sure all elements meet the technical requirements. ASM staff will contact you if anything needs to be revised before copyediting and production can begin. Otherwise, you will be notified when your proofs are ready to be viewed.

Sincerely,
Jan Claesen
Editor
Microbiology Spectrum